# Seasonal Aridity in the Indo-Pacific Warm Pool During the Lateglacial Driven by El Niño-like Conditions

Petter L. Hällberg[1], Frederik Schenk[1,2], Kweku A. Yamoah[3], Xueyuen Kuang[4], Rienk H. Smittenberg[1]

[1]Department of Geological Sciences and Bolin Centre for Climate Research, Stockholm University, 106 91, Sweden,
[2]Rossby Centre, Swedish Meteorological and Hydrological Institute, 601 76, Sweden,
[3]BioArc, Department of Archaeology, University of York, York, YO10 5DD, UK
[4]School of Atmospheric Sciences, Nanjing University, 210023, China

*Correspondence to*: Petter L. Hällberg (petter.hallberg@geo.su.se)

**Abstract.** Island South-East Asia (ISEA) is a highly humid region that hosts the world's largest tropical peat deposits. Most of this peat accumulated only relatively recently during the Holocene, suggesting that the climate was drier and/or more seasonal during earlier times. Although there is evidence for savanna expansion and drier conditions during the Last Glacial Maximum (LGM, 21 ka BP), the mechanisms behind hydroclimatic changes during the ensuing deglacial period has received much less attention and are poorly understood. Here we use CESM1 climate model simulations to investigate the key drivers behind ISEA climate at the end of the Lateglacial (14.7-11.7 ka BP), with a focus on the last stadial the Younger Dryas (12 ka BP). We further simulate the preceding Allerød (13 ka BP) interstadial climate and perform a sensitivity experiment to disentangle the climate impacts due to orbital forcing and Lateglacial boundary conditions against a slowdown of the Atlantic Meridional Overturning Circulation (AMOC). A transient simulation (TRACE) is used to track the climate seasonality and orbitally driven change over time during the deglaciation into the Holocene. In agreement with proxy-evidence, CESM1 simulates overall drier conditions during the Younger Dryas and Allerød. More importantly, ISEA experienced extreme seasonal aridity, in stark contrast to the ever-wet modern climate. We identify that the simulated drying and enhanced seasonality in the Lateglacial is mainly the result of a combination of three factors: 1) large orbital insolation difference on the northern hemisphere (NH) between summer and winter, in contrast to the LGM and the present day; 2) a stronger (dry) East Asian winter monsoon caused by a larger meridional thermal gradient; and 3) a major reorganization of the Indo-Pacific Walker Circulation with an inverted land-sea circulation with a complete breakdown of deep convection over ISEA in NH winters. The altered atmospheric circulation, sea surface temperature and sea level pressure patterns led to conditions resembling extreme El Niño events in the modern climate and a dissolution of the Inter-Tropical Convergence Zone (ITCZ) over the region. From these results we infer that terrestrial cooling of ISEA and at least a seasonal reversal of land-sea circulation likely played a major role in delaying tropical peat formation until at least the onset of the Holocene period. Our results also suggest that centennial to millennial shifts in AMOC strength modifies the Pacific Ocean hydroclimate via alteration of the position of the ITCZ, and a modulation of the Pacific Walker Circulation. However, Lateglacial AMOC shifts are overall less important than hydroclimate changes due to orbital forcing and boundary condition changes relative to the modern climate.

# 1 Introduction

The Indo-Pacific Warm Pool (IPWP) is the world's largest region with sea-surface temperatures permanently above 28 °C. This 'heat and steam engine of the globe', is a major source of atmospheric deep convection and heat flux in the tropics and plays a key role for monsoon systems that directly affect approximately half of the global population (De Deckker, 2016; Chabangborn et al., 2018). Currently, the IPWP is undergoing rapid change and has doubled in size during 1981-2018 compared to 1900-1980, and the growth rate is accelerating (Roxy et al., 2019). The IPWP region is highly interconnected to interhemispheric changes through latitudinal shifts of the Intertropical Convergence Zone (ITCZ) and monsoon systems as well as to zonal variations of the Walker Circulation, in particular the El Niño Southern Oscillation (ENSO) over the Pacific Ocean. Located in the central IPWP, Island South-East Asia (ISEA) is wet year-around, hosting tropical rainforests and peat swamp forests (Beck et al., 2018; Malhi and Wright, 2004). Based on climate simulations, it is very likely that rainfall variability will increase in the future under an expanding IPWP and enhanced ENSO variability (Grothe et al., 2020; IPCC, 2021; Wengel et al., 2021). Increasing rain variability in the context of amplified ENSO modes may enhance the risk for seasonal droughts and forest fires in today's ever-wet tropical rain forests. These changes may further weaken the small hydrological cycle of these ecosystems potentially turning carbon sinks into carbon sources under global warming. In this context, it is noteworthy that most of the peatlands and rain forests of ISEA are rather young and became established only during the early to mid-Holocene (Dommain et al., 2014). Whether this rather young age is due to a generally drier climate prior to the Holocene, or rather due to changes in precipitation variability or seasonality, remains a challenging research question.

There is also much debate about the past variability and states of ENSO. Both modelling and proxy evidence indicate weakened ENSO variability and "La Niña-like" conditions with a strong zonal sea surface temperature (SST) gradient during the mid-Holocene, prior to an increase in ENSO variability towards the near-present state in the Late Holocene (Brown et al., 2020; Carré et al., 2021; Chen et al., 2016; Emile-Geay et al., 2016; Grothe et al., 2020). However, the state of ENSO is not as well constrained during the Lateglacial (~14.7-11.7 ka BP) and the LGM (~21 ka BP). There is evidence for increased ENSO variability during both the LGM and Lateglacial, under El-Niño like states in terms of weaker Pacific SST gradients and larger $\delta^{18}O$ variability in foraminifera (Clement et al., 1999; Koutavas and Joanides, 2012; Sadekov et al., 2013), but these interpretations have been challenged. Instead, a strengthened seasonal cycle (Ford et al., 2015; Zhu et al., 2017) and a supressed interannual ENSO variability has been proposed, being consistent with both proxies and model simulations (Ford et al., 2015; Leduc et al., 2009; Liu et al., 2014a; Zhu et al., 2017). A recent PMIP4 climate model ensemble indicates weakened ENSO variability at the LGM, but there is a large spread between different models (Brown et al., 2020).

Reconstructions from speleothems and other proxies (Dang et al., 2020; Konecky et al., 2016; Russell et al., 2014) indicate widespread drier conditions in the IPWP before the early Holocene (*ca.* 10-11 ka BP). Drier overall conditions together with changes in seasonality and ENSO variability suggest that large-scale atmospheric mechanisms were very different during the

Lateglacial. However, proxy data typically provide a multi-annual integrated signal that does not allow to disentangle general climatic trends from the influence of changes on seasonal time-scales.

On a local to regional level, a number of pollen and biomarker-based proxy records, as well as floral and faunal records indicate open grassland/shrub vegetation and drier conditions during the LGM and the subsequent deglacial period (referring to the period following the end of the LGM until the end of the Younger Dryas). This has sparked a debate about to what extent savanna expanded in ISEA during (potentially seasonally) drier conditions, and a 'Savanna Corridor' hypothesis outlining a savanna biome across the exposed Sunda shelf has been proposed (Bird et al., 2005; Heaney, 1991; Louys and

Roberts, 2020; Wurster et al., 2010, 2019) (Fig. 1). However, evidence remains inconclusive for key regions, in part because much of the proposed savanna areas were inundated during deglacial 120 m sea level rise (Hanebuth et al., 2011). Whether the proposed savanna formed a contiguous "corridor", or if the savanna vegetation was interrupted by a rainforest between Sumatra and Borneo, remains an open question (Cannon et al., 2009; Wurster et al., 2019). The existence of a savanna corridor during the LGM has previously been questioned by Raes et al. (2014) based on rainforest species distribution

modelling forced by two climate models. These showed increased precipitation compared to modern climate on north Borneo and on the Sunda shelf (Brady et al., 2013). However, the simulated increase in precipitation is at odds with both a PMIP3 model synthesis of eleven climate model simulations (PMIP3 Synthesis maps, 2014; https://pmip3.lsce.ipsl.fr/ accessed 13-06-2022), and with cave deposit $\delta^{18}O$ records in that area (Partin et al., 2007; Wurster et al., 2010), suggesting that their results may be based on a too wet climate model output. The lack of proxy records and a limited understanding

about the driving mechanisms for an altered climate state during the deglaciation call for the need of climate modelling experiments to unravel drivers behind such drastic changes.

Although it is well accepted that the Sunda climate was generally drier and colder prior to the Holocene, previous research has mainly focused on the LGM, with much less attention on transient changes during the subsequent deglaciation. Also,

how the generally drier climate might have been related to seasonality changes rather than just mean changes in precipitation, has received little attention. The goal of this study is to investigate the seasonal climate evolution in the Sunda region by analyzing climate model simulations, a transient coupled ocean-atmosphere model, and comparing it to available proxy data. We focus here on the period known as the Younger Dryas, representing the very end of the Lateglacial before the start of the warm and wet Holocene. This enables further insights into the deglacial climate, in a period with different

seasonal orbital forcing compared to both the pre-industrial (PI) and the more studied LGM. In particular, we quantify the length and aridity of the dry season, since these parameters can have a much stronger effect on ecosystems than the total annual precipitation (Malhi and Wright, 2004). We then investigate the large-scale forcings and mechanisms that dominate the deglacial tropical climate on seasonal time scales in comparison to the modern pre-industrial climate.


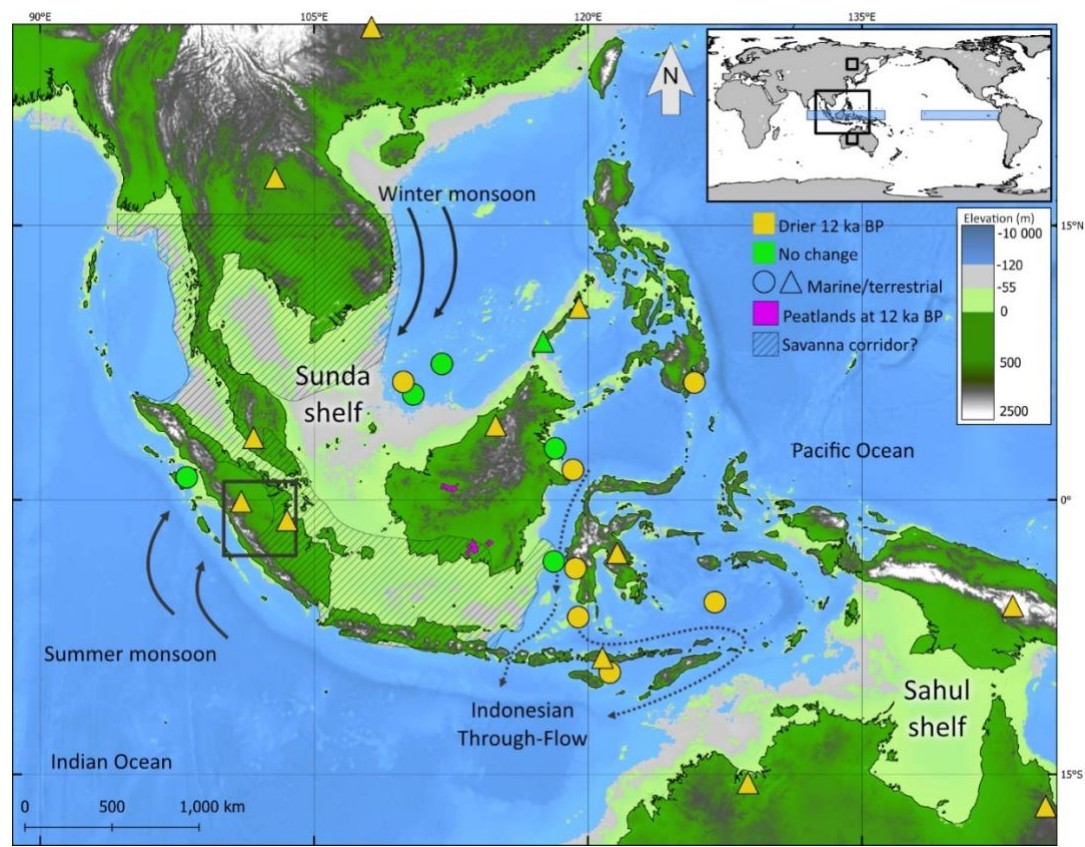

Figure 1: Topographic map of South-East Asia and a compilation of available climate proxy records (supplementary Table 1). Yellow markers indicate drier or more seasonal climate at 12 ka BP compared to PI, and green indicates no notable difference. Purple areas denote the peatland extent at 12 ka BP (Dommain et al., 2014). Exposed land due to lower sea level at 12 ka BP shown in light green and in light gray for the LGM. The hatched area indicates the proposed savanna/seasonal deciduous forest corridor on the Sunda shelf at the LGM (Cannon et al., 2009). Black square on the regional map indicates the area used to track long-term changes in seasonality from TRACE (Fig. 2 f, g). Small black squares in the global map inset show the areas used for estimating the thermal gradient over ISEA and blue squares are used for Walker Circulation Index calculations (Fig. 7a).

## 2. Materials and methods

### 2.1 Proxy data compilation

To synthesize the available proxy-evidence for the IPWP region at the end of the Lateglacial, we compiled a range of paleoclimate proxy records based on pollen assemblages and stable isotopes, spanning at least the last 12 kyr (see Table S1). We follow the interpretations of the authors, where $\delta^{13}C$ of leaf waxes is considered to reflect C3/C4 vegetation abundance influenced by aridity or climate seasonality, and where $\delta^{18}O$ and $\delta D$ mainly reflects precipitation amounts. Speleothem $\delta^{18}O$ records in the Sunda region from Borneo at 4° N (Buckingham et al., 2022; Partin et al., 2007), equatorial Sumatra at 0° N (Wurtzel et al., 2018) and Java at 8° S (Ayliffe et al., 2013; Griffiths et al., 2009) show the overall deglacial transient climate evolution, but also that regional climate shifts are partly out of phase between the northern and southern hemisphere due to

bipolar seesaw mechanisms and the impact from different monsoonal systems (Ayliffe et al., 2013). Greenland ice $\delta^{18}O$ (NGRIP members, 2004) is shown as a comparison to the Sunda speleothem records to highlight that the deglacial climate

transition is similar in terms of timing and direction at polar latitudes and Sunda's tropical latitudes. The topographic map data used to outline the modern, 12 ka BP and LGM coastlines in figures was derived from NOAA's 1-arc minute ETOPO1 present-day elevation dataset (Amante and Eakins, 2009). 12 ka BP sea level was set to -55 m, and LGM sea level was set to -120 m compared to current sea level.

## 2.2 Climate model simulations

To investigate the transient change in temperature and precipitation seasonality from the LGM to the present on Sunda, the TraCE21ka simulation data (He, 2011; He et al., 2013; Liu et al., 2009) (hereafter referred to as TRACE) is used. The TRACE simulation was performed with the CCSM3 model developed by NCAR, and is a full-complexity coupled atmosphere-ocean model. For the transient paleoclimate runs, CCSM3 was forced with changes in greenhouse gas concentrations and orbital insolation changes. In addition, continental ice sheets and sea level changes were adjusted over

time and the fresh water forcing from melting ice sheets and/or drainage of proglacial lakes was set as described in detail by (He, 2011). The spatial model resolution is T31_gx3, (3.75° x 3.75° in the atmosphere, ~415 km). Here, we analyzed TRACE data in the region from present-day Sumatra, Indonesia, which represents the central Sunda region. We select a location (3° S–1° N, 100° E–104° E) that is currently land surface to avoid signal variation arising from changing the model surface type from water to land during the deglacial transgression.


Because the transient simulation with TRACE had to be performed at a rather coarse spatial resolution, we additionally analyze higher resolution snapshot simulations for the Younger Dryas and Allerød intervals that better represent the regional exposure of today's inundated Sunda shelf. We additionally present results from a "cold-ocean-only" sensitivity experiment using the boundary conditions from Allerød, but with the cold ocean state from Younger Dryas, mimicking a freshwater

hosing experiment from higher input of high latitude meltwater (20 m/kyr at 12 ka BP and 5 m/kyr at 13 ka BP) and weaker Atlantic Meridional Overturning Circulation (AMOC) (He, 2011) prescribed from TRACE. The model setup and boundary conditions used for the higher resolution CESM1.0.5 model simulations of 12.17 ka BP (hereafter referred to as 12 ka BP), 13 ka BP and 13 ka BP with 12 ka BP ocean state (hereafter referred to as 13kYD) have been previously described by Schenk et al. (2018). CESM1 is a newer version of CCSM3 that was used previously for the transient simulations of

TRACE. The CESM1 simulation here is based on a coupled atmosphere-land-sea ice model, with sea surface temperature and sea ice fraction prescribed from TRACE. The CESM1 simulation has a horizontal resolution of 0.9° x 1.25° (~120 km). Continental ice sheets and corresponding sea level and hence land-sea distribution in CESM1 are based on the ice sheet reconstruction from GLAC-1B (Schenk et al., 2018), a precursor of the PMIP4 ice sheet reconstruction (GLAC-1D: Ivanovic et al., 2016). The vegetation cover in the 12 ka BP simulation was prescribed from the pre-industrial state of CESM1, and

kept constant during the simulation. The vegetation on exposed shelf areas was set by next-neighbor interpolation to reflect adjacent land areas. To avoid some inconsistencies due to changing surface types when interpolating CESM1 model output onto ancient coast lines, model grid points that were partly sea and partly land, i.e., the coastlines, were masked out from the 12 ka BP simulation and filled using nearest neighbor interpolation. The coastlines in the 12 ka BP and PI simulations are shown in Fig. 3.

**2.3 Analysis**

To compare transient seasonality changes in temperature and precipitation simulated by TRACE, we used insolation data for the central Sunda shelf (2° N) from 20 ka BP to the present (Laskar et al., 2004). To quantify changes and severity of precipitation seasonality, we used <60mm precipitation to define a dry month under tropical conditions following Köppen's classification (Beck et al., 2018). Raes et al. (2014) and Malhi and Wright (2004) used a 100 mm/month classification, based

on the fact that 100 mm is close to the monthly evaporation in present-day rainforests. Since 12 ka BP was colder, leading to lower potential for evapotranspiration, we decided to use the lower value of 60 mm/month. Given the strong simulated changes, the results are insensitive to the exact definition. As a first order approximation, we estimate the potential thermodynamic contribution of SST cooling on the simulated reductions in total precipitation by calculating a 7 % decrease in total precipitation per Kelvin cooling, based on the Clausius-Clapeyron relation between changes in temperature and

potential water vapor uptake by the atmosphere. We then compare this to the simulated change in total precipitation by CESM1 to separate the theoretical thermodynamic contribution from dynamical changes. The temperature gradient in the Asian-Australian monsoon system, responsible for driving the north-south potential strength of the monsoons, was derived between 120-130° E, 20-30° S for Australia, and 120-130° E, 50-60° N for Siberia (boxes on Fig. 1).

We calculated a Walker Circulation Index (WCI) to quantify the strength of the Pacific Walker Circulation from monthly mean sea level pressure as the difference in pressure in hPa between the IPWP and West Pacific (80° E to 160° E, 5° S to 5° N) and the East Pacific (160° W to 80° W, 5° S to 5° N), boxes shown in Fig. 1, following Kang et al. (2020). To compare the simulated WCI during PI and 12 ka BP with observational data, the ERA5 dataset (Hersbach et al., 2020) was used for monthly mean sea level pressure data from 1979 to 2020 to calculate observed changes in the Walker Circulation Index. The

average WCI during extreme El Niño years (1983, 1998 and 2015) were compared to the CESM1 simulations. ERA5 SST data was used from the same El Niño events to compare the SST anomalies during El Niño events for January, which usually is the month with the highest El Niño strength. January SST anomalies in ERA5 data was calculated as the January mean of the entire ERA5 dataset (1979-2020) subtracted from the average January SST during the El Niño events in 1983, 1998 and 2015. The simulated SST anomaly at 12 ka BP compared to PI was calculated as the difference in SST between 12 ka BP

and PI, with the average tropical (23.5° S to 23.5° N) SST cooling of 2.08 °C between the periods subtracted. This was done to show the relative temperature change within the Pacific Ocean at 12 ka BP.

**2.4 Model validation**

The CESM1 version used here is one of the scientifically validated releases by NCAR and is widely used in different studies
including a validation of monsoon characteristics (Meehl et al., 2020). To further validate the performance of our CESM1 simulations over the IPWP region, we simulated the present day (~year 2000) climate state using the CESM1 F2000 climo component set with averaged HadOIBl ocean data of 1982-2001 (Hurrell et al., 2008) for 40 years, and compared the mean outputs with the ERA5 1979-2020 reanalysis dataset. Shown in Fig. S1 is the annual difference between the simulated and observed precipitation and temperature. Fig S1a-d show the ISEA patterns in precipitation and temperature, while Fig S1e-f
show the simulation biases over the whole Indo-Pacific region. The main discrepancies in the CESM1 simulations of modern climate compared to observations are 1) a westward displacement of the precipitation maximum over the Indian Ocean, 2) the well-known double ITCZ over the Pacific (Zhang et al., 2019), and 3) a too strong Pacific Cold tongue in CESM1 simulations with a dynamic ocean (Meehl et al., 2020). These biases have been further discussed in the context of monsoons and ENSO by Meehl et al. (2020). The Maritime Continent precipitation is well reproduced by CESM1 (Fig. S1 a, b, e); discrepancies
consist of a slightly too wet peninsular Malaysia and southern Indonesia. The complex land-sea distribution in the Maritime Continent, with high topography on small islands in a mostly marine setting, leads to large differences in temperature, convection and orographic effects over small spatial distances, and can result in differences between models as seen for example by Kageyama et al. (2021). Temperature and precipitation biases on islands with steep topography are both negative and positive in CESM1 (Fig. S1 e,f), and are mostly associated with the lower resolution of CESM1 compared to ERA5;
observations are highly variable in steep terrain, which are averaged over several grid points and thus not represented in the simulation, in particular for mountain ranges on western Sumatran, Borneo and Papua New Guinea. The bias reported by NCAR (https://www.cesm.ucar.edu/experiments/cesm1.0/diagnostics/b40.1850.track1.1deg.006/atm_863-892-obs/ accessed 13-06-2022), Meehl et al. (2020) and Fig. S1 specifically for CESM1 and more generally for PMIP models (Brown et al., 2020; Kageyama et al., 2021) highlights the need to validate paleo-simulations against available proxy data when possible.

**3 Proxy synthesis in the Sunda region**

The transition from Lateglacial into the current interglacial of the Holocene is marked by drastic climatological changes (Ayliffe et al., 2013; Griffiths et al., 2009; Partin et al., 2007; Wurtzel et al., 2018) relative to that of the present day (Fig. 2). While ISEA is wet year-around in the modern climate because of its location in the IPWP, this was not the case before the early Holocene. Based on previous proxy studies from the ISEA/IPWP region, the general deglacial climate sequence
consists of a cold and dry LGM, followed by further drying during Heinrich Stadial 1 (HS1) (17-14.7 ka BP), a warmer and more humid Bølling-Allerød (BA) (14.7-12.8 ka BP), reverting back into the cold and dry Younger Dryas (12.8-11.7 ka BP), leading into the considerably wetter and warmer Holocene (11.7-0 ka BP) (Fig. 2) (Ayliffe et al., 2013; Buckingham et al., 2022; Partin et al., 2007; Wurtzel et al., 2018). This trend is similar to ice core records on Greenland, indicating that

teleconnections between high latitude and tropical climates are important (Yuan et al., 2018). The timing of the driest period

is different for records south of the equator, where the southward shifted ITCZ during North Hemisphere (NH) cold events

resulted in enhanced Australian-Indonesian summer monsoon precipitation, with relative wetting on the southern hemisphere

during Younger Dryas and HS1 (Ayliffe et al., 2013) (Fig. 2). The general deglacial sequence is however similar, and

overall, the implication is that the Younger Dryas studied here is somewhat less extreme than the LGM but still very distinct

from the Holocene.


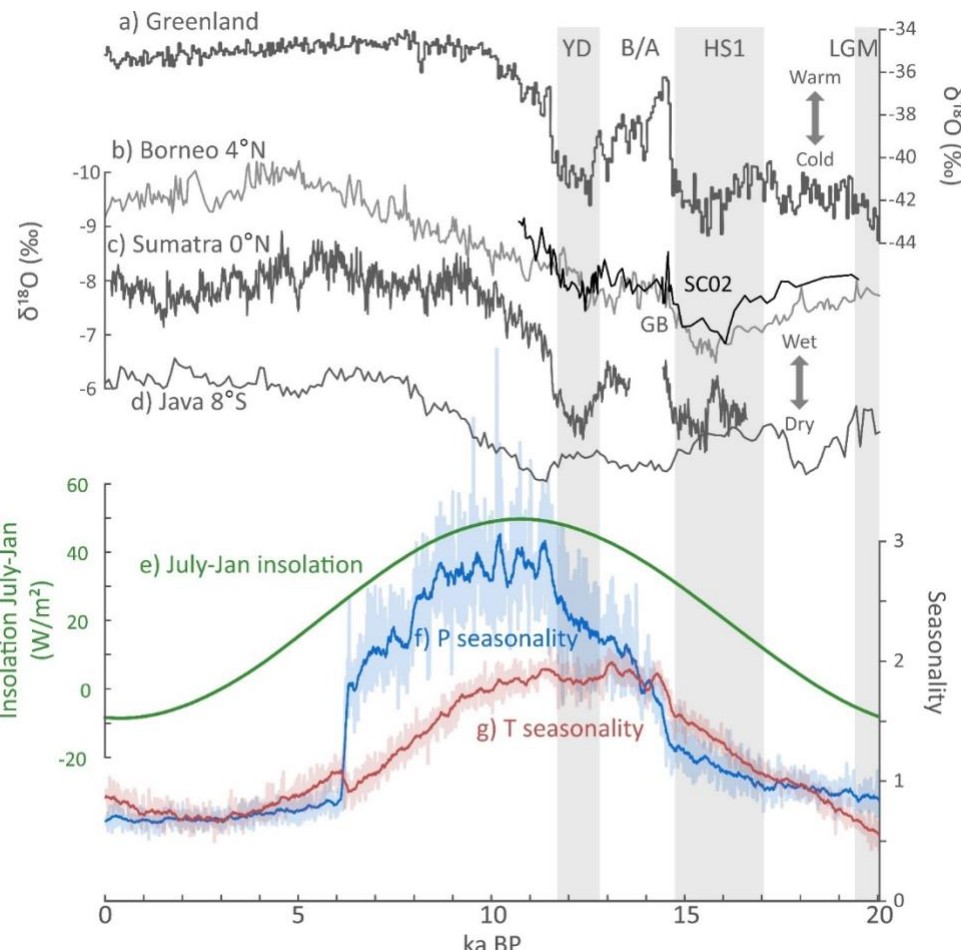

Figure 2: Long-term transient changes in temperature and hydroclimate based on proxy evidence in comparison to seasonality changes in

terms of orbital forcing (2° N) and simulated precipitation (P) and temperature (T). a) Greenland (NGRIP) ice core $\delta^{18}O$ indicative of
northern hemisphere high latitude temperature (NGRIP members, 2004). b) speleothem $\delta^{18}O$, indicative of precipitation amount, from
Borneo (Partin et al., 2007: GB, grey), and (Buckingham et al., 2022: SC02, black); c) Sumatra (Wurtzel et al., 2018); and d) Java
(Ayliffe et al., 2013; Griffiths et al., 2009). e) Insolation difference between NH winter and summer calculated from July minus January
insolation from 2 °N (Laskar et al., 2004). Seasonality is calculated as JJA/DJF for precipitation (f, blue) and JJA-DJF for temperature (g,

red) based on the TRACE simulation (He, 2011) over Sunda (3° S-1° N, 100°-104° E), with 19 point running mean in bright colors and
each data point in pale color. The strong changes in seasonality closely follow the orbital forcing seasonality which is evident in climate
model simulations but not visible in the seasonally integrated proxy-signals.

Focusing on 12 ka BP specifically, a compilation of climate and environmental proxy records based on stable isotopes and pollen indicate widespread drier conditions in the Sunda area at 12 ka BP compared to PI in the region (Fig. 1). However, local deviations are found in some records that do not record a significant change between modern and deglacial climates, notably around Borneo which appears to have remained relatively wet according to proxies (Fig. 1) and climate simulations (Brady et al., 2013). In contrast to a nearby terrestrial speleothem record on Sumatra, Niedermeier et al. (2014) did not find a clear difference over the LGM-deglacial-Holocene period in their record based on terrestrial biomarkers found in a marine record off Sumatra. They attribute this difference to local variations in topography and a different response to sea level rise. Pollen and biomarker records in the southern South China Sea, which mainly receives input from Borneo (Yang et al., 2020), indicate that only small changes in humidity occurred on the Sunda shelf from the LGM to the present (Chabangborn et al., 2018; Hu et al., 2003, 2002; Wang et al., 2007). On the contrary, Yang et al. (2020) found evidence in a nearby marine core for a precipitation minimum during meltwater pulse MWP-1A at ~14 ka, and a subsequent slow increase towards the Holocene.

The synthesis of local records suggests an overall drier and/or seasonal climate during the deglacial period including 12 ka BP, but with some more local ever-wet exceptions, particularly around Borneo. In line with this, large-scale peat formation in ISEA, which requires ever-wet or water-logged conditions, first initiated in central Borneo (purple in Fig. 1) around 14 ka BP (Dommain et al., 2014). In contrast, almost all other peatlands in the region started to accumulate only relatively recently, during the Holocene, in particular following the mid-Holocene sea level high stand of +5 m a.s.l. at 5000 yr BP (Dommain et al., 2014). This indicates that the continuously anoxic, waterlogged conditions required for peat accumulation did not widely occur before that time, implying that (ancient) land areas of ISEA must have experienced significant dryness, at least seasonally, in today's ever wet region with exception of Borneo.

## 4 Model simulations

### 4.1 Transient climate evolution since the LGM

Although the transient simulation of annual precipitation over the ISEA region agrees with the overall drier conditions during the deglaciation reflected in proxy-records (Fig. 1), a key factor appears to be an uneven seasonal distribution of total precipitation compared to the present (and to some extent compared also to the LGM) (Fig. 2f). Such a seasonality signal is usually not directly detectable from proxy records. Contrasting the seasonal cycle as difference between summer (JJA) and winter (DJF) climate states, the transient TRACE simulation indicates a major shift to a more seasonal climate on central Sunda during the deglaciation and early Holocene, both in terms of precipitation and temperature (Fig. 2f-2g). Throughout this paper, 'summer' and 'winter' refer to the northern hemisphere seasons, unless specified differently. At the present day, the temperature is stable over the year with generally <1 °C difference between summer and winter. Precipitation is slightly lower in summer than winter resulting in a precipitation ratio for JJA/DJF near or below 1 (Fig. 2f). At 12 ka BP however,

the temperature seasonality is doubled to ~2 °C, and precipitation seasonality is completely reversed, with a much drier DJF and the majority of precipitation falling in NH summer, resulting in JJA/DJF ratio of 2 to 3. This rainfall seasonality strongly follows insolation (orbital precession, the 21 ka cycle), which has previously been observed in Sunda region environmental proxies (Partin et al., 2007; Wurster et al., 2019). The insolation difference between the NH summer and winter at 2° N was

at a maximum at 11 ka BP, with a 49.7 W/m$^2$ (~13.2 %) stronger insolation in July. Currently, July insolation is 8.4 W/m$^2$ (2 %) lower than during January (Fig. 2e). The rapid increase in climate seasonality at the end of the Younger Dryas according to TRACE (Fig. 2f) is primarily related to an increase in summer precipitation when the early Holocene warming and Sunda shelf inundation commenced, and winters remained cold and dry during the NH winter insolation minimum.

TRACE seasonality on Sunda at 12 ka BP is also much higher compared to LGM. An important difference between the LGM and 12 ka BP is that the insolation difference between NH winter and summer was near a minimum at the LGM and PI, while it was near maximum at 12 ka BP (Fig. 2e). These seasonal orbital forcing extremes provide an important mechanism for an increase in seasonality towards 12 ka BP and the early Holocene relative to LGM and PI. This implies that a mere focus on annually integrated changes (as reflected by most proxy-records; Bova et al., 2021; Liu et al., 2014), will

only yield an incomplete explanation for the drastic paleo-environmental changes. We further evaluate the 12 ka BP climate state by analyzing a higher resolution simulation.

## 4.2 Highly seasonal hydroclimate in the Sunda region at 12 ka BP

To gain a more realistic representation of topography, sea level changes and atmospheric deep convection over tropical regions, model simulations focused on 12 ka BP and PI climate states were done using CESM1.0.5 (Schenk et al., 2018).

The simulation of 12 ka BP generally confirms the proxy-evidence for widespread drier conditions (Fig. 3) as a criterion for the absence of tropical rainforest and peat in favor of savanna in the ISEA/IPWP region. Simulated precipitation for PI shows that most precipitation falls over ISEA land areas, but precipitation over land is strongly reduced at 12 ka BP and shifted to areas over the Indian and Pacific Oceans. The higher resolution CESM1 precipitation changes are in general agreement with TRACE but show a more pronounced drying over land (comparison not shown).


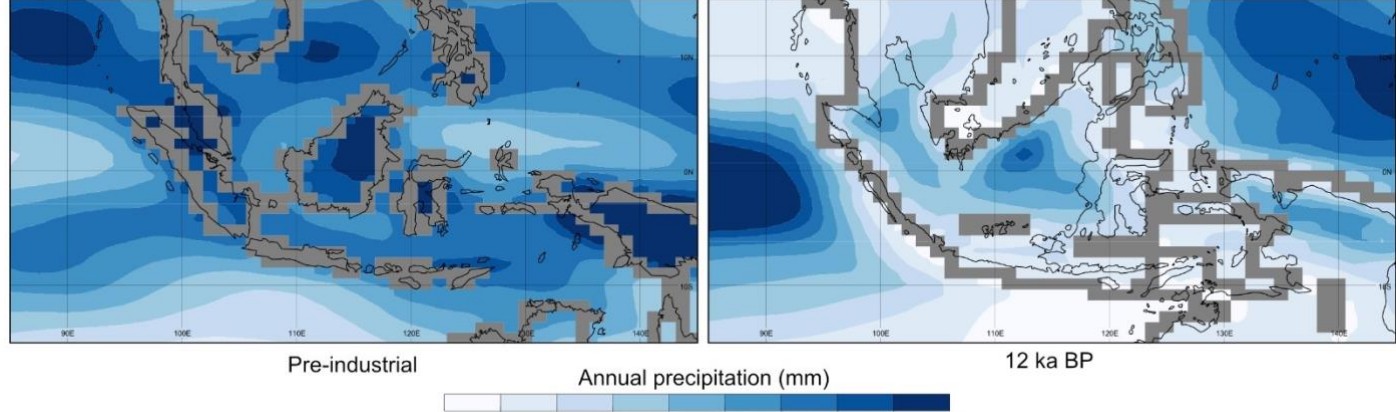

Figure 3. Comparison of annual mean total precipitation (mm/year) at PI and 12 ka BP based on CESM1. The generally drier conditions are consistent with generally drier proxy-evidence during the Lateglacial (e.g., Fig. 2b-d). Note that precipitation at 12 ka BP shifts from the land onto the sea. Seasonal precipitation is shown in Fig. S2 with almost no precipitation over most ISEA areas in NH winter. Black lines are coastlines for each period based on present day bathymetry (0 m elevation for PI, -55 m for 12 ka BP), and greyed out areas are partly land and partly sea in the CESM1 simulations, i.e., model coastlines.

Moreover, our simulations show that the climate was not only generally drier at 12 ka BP in the Sunda region, but extremely seasonal, in stark contrast to the PI ever-wet conditions. Figure S2 shows that while all months are generally drier at 12 ka BP than in PI, the most substantial drying occurs in NH winters (DJF). By calculating the number of dry months under tropical conditions (defined as P <60 mm/month) for each grid point, we find that seasonal droughts were occurring almost everywhere in ISEA at 12 ka BP according to our CESM1 simulation (Fig. 4). Consistent with proxy-evidence (Fig. 1) and TRACE (Fig. 2), this seasonal dryness at 12 ka BP is very different to the constantly wet PI climate which the entire ISEA and southern mainland SE Asia experiences (Fig. 4 inset). Only central Borneo remained wet throughout the year at 12 ka BP indicating an exceptional agreement of the simulated spatial precipitation change with proxy evidence and peatland initiation around Borneo (Fig. 1). Apart from Borneo, the dry period duration was approximately 3-4 months in most locations around the equator, 5-7 months for mainland ISEA, and as much as 8-10 months for the Java and the Sahul shelf. That Borneo and the eastern ISEA receive relatively more precipitation, while the rest of Sunda becomes dry during periods of freshwater input into the North Atlantic, as was the case around 12 ka BP, is also supported by freshwater hosing simulations (Mohtadi et al., 2014).

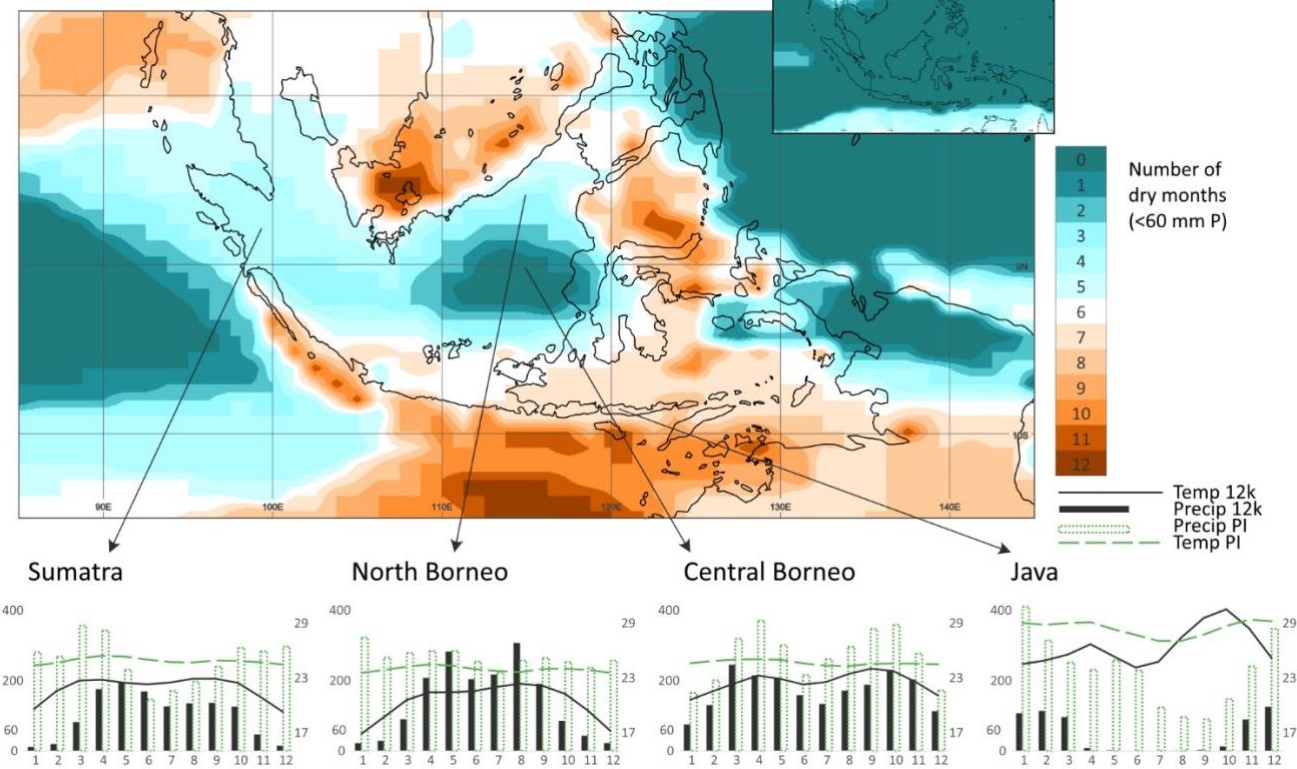

Figure 4: Number of simulated dry months per year at 12 ka BP and comparison of the regional annual temperature and precipitation cycles for 12 ka BP and PI. The inset shows the number of dry months (<60 mm P/month) per year for PI. Climate diagrams show the monthly average temperature and precipitation from PI (green) and 12 ka BP (black) based on CESM1 simulations. Sumatra, Java and North Borneo locations approximately correspond to speleothem cave locations from the reconstructions in Fig. 2.

Drastically reduced temperatures are simulated in particular for the exposed, low lying Sunda shelf east of Malaysia and Sumatra, as well as in the highlands of northern Borneo (Fig. S3 and S4). In those areas, surface temperature was down to ~18 °C in January at 12 ka BP compared to PI temperatures of 25-27 °C in most low-lying areas and around 22 °C in the highlands (Fig. S4). Simulated annual mean ocean temperatures were ~2 °C cooler at 12 ka BP, which is similar to the deglacial cooling proposed in a recent compilation of SST's and thermocline temperatures in ISEA (Dang et al., 2020), but the warming starts earlier according to the proxies. The ocean temperature proxies show peak warmth already at ~10 ka BP, and only ~0.5 °C cooler temperature at 12 ka BP compared to present. This early Holocene warming is inconsistent with most climate models and has been proposed to be related to a seasonal bias of marine proxies known as the 'Holocene Temperature Conundrum' (Bova et al., 2021; Liu et al., 2014b). The simulated extreme seasonality changes over tropical land areas provide further evidence for the need to distinguish mean annual changes from the potentially more dominant role of changing seasonality – in particular seasonal aridity.

Lower IPWP SST can account for part of the reduced precipitation at 12 ka BP, since a theoretical ~7 % decrease in humidity and subsequent convective precipitation from the ocean can be expected per Kelvin cooling (Clausius-Clapeyron relation). A decrease of around 10-30 % precipitation in January can thus be attributed to the cooling, which falls much short of the total simulated decrease of 50 to 99 % (Fig. S5). We thus conclude that the drastic reduction of winter precipitation compared to PI requires fundamental large-scale changes in terms of atmospheric circulation and cannot be explained alone

by a lower moisture availability due to cooling at 12 ka BP.

### 4.3 Mechanisms driving strong seasonality in the Sunda region at 12 ka BP

As outlined in Fig. 2, transient seasonal insolation changes serve as a good general explanation for seasonal temperature and precipitation changes over ISEA. However, additional changes are required to explain the drastic reductions in seasonal precipitation given the relatively small ocean cooling signal that is insufficient to explain the drastic moisture changes. A key

change on hemispheric scale is that mid to high NH latitudes cooled much more than the tropics at 12 ka BP compared to PI (Fig. S6). This is due to the remaining presence of ice-sheets and sea ice, and a weaker AMOC (He, 2011). NH cooling was then enhanced further in particular during winters due to low orbital winter insolation. Thus, the pole-equator thermal gradient between the IPWP low- and Siberian high-pressure systems, driving the monsoons, was much larger. CESM1 simulates a ~27 % larger (winter) thermal gradient at 12 ka BP compared to PI between the large land areas of Siberia and

Australia ($\Delta T = 76$ °C, compared to 60 °C at PI). As a result, CESM1 simulates a much stronger East Asian winter monsoon at 12 ka BP compared to PI (Fig. 5). Such a strengthened East Asian winter monsoon is also supported by proxy evidence (Griffiths et al., 2009) and stronger temperature gradients in the South China Sea during the Lateglacial, both according to SST proxies and the TRACE simulation (Huang et al., 2011; Wen et al., 2016).

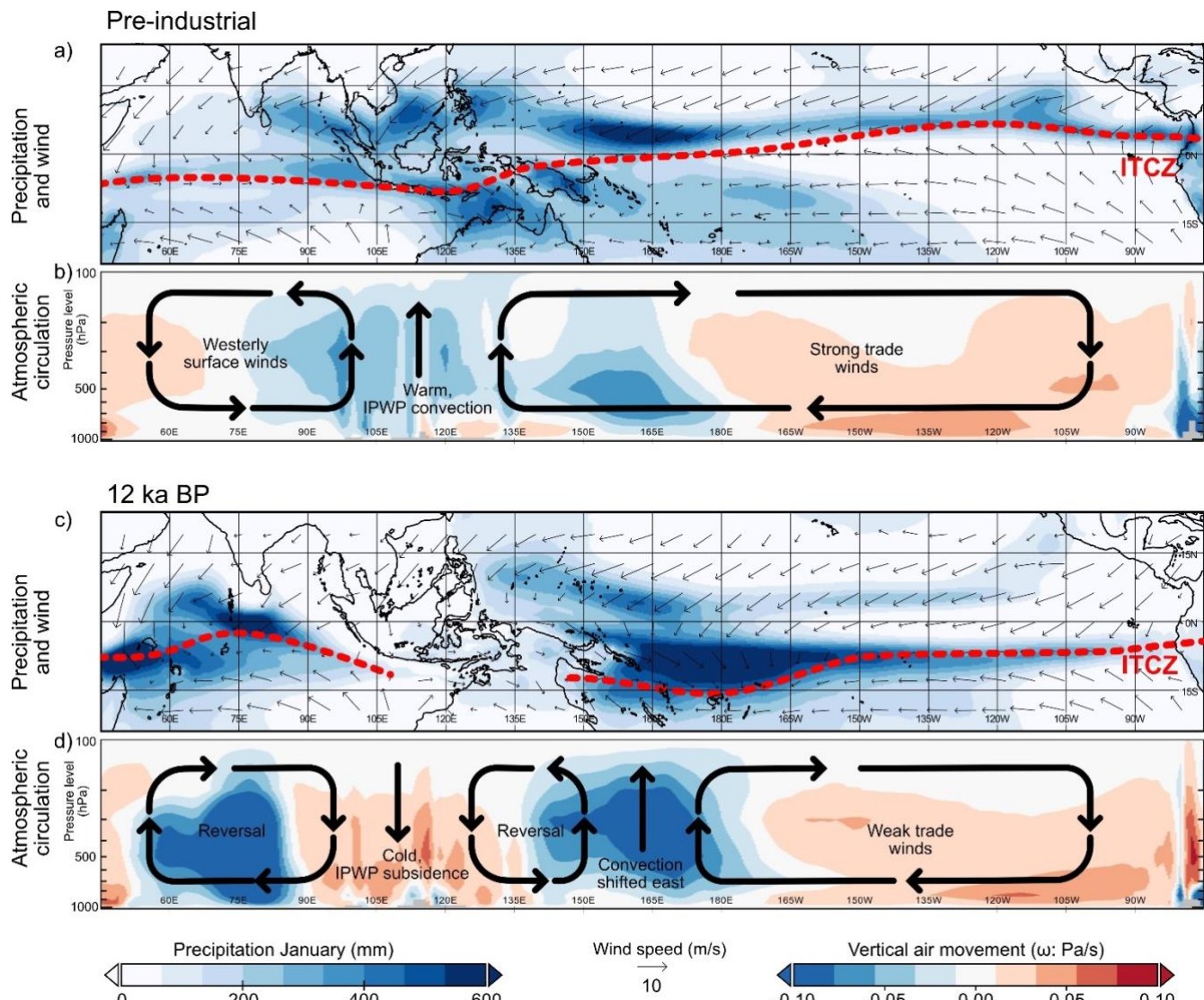

Figure 5. Simplified mechanistic explanation for January precipitation patterns during the pre-industrial (top) and 12 ka BP (bottom) in terms of horizontal and vertical wind patterns. a) and c) show mean total precipitation (mm/month) and surface wind vectors (m/s) for January; b) and d) Omega parameter (ω: Pa/s) representing vertical air movement (ω<0, rising in blue; ω>0 sinking in red) along the equator over the Indo-Pacific region. The approximate January location of the ITCZ and atmospheric Walker Circulation patterns are indicated as red lines (a, c) and black arrows (b, d), respectively. Note that the atmospheric circulation cells are inverted over the IPWP region at 12 ka BP where subsiding air masses, a breakdown of deep convection and divergent winds from land to sea cause seasonal aridity over tropical land areas. For details on the weakening and/or reversal of the zonal winds, see Fig. S7.

For the pre-industrial simulation, the ITCZ rain bands are present over ISEA (Fig. 5a) throughout the year (annual and seasonal precipitation for both simulations are shown in Fig. 3 and Fig. S2, respectively). Precipitation during NH winters is drastically reduced at 12 ka BP with a majority of Sunda receiving <40 mm precipitation compared to generally 200-500 mm for PI (Fig. 5) during the driest month. The driest month occurs in January for most locations except Java and Oceania which

is driest during July (during the Australian-Indonesian winter monsoon). The rain bands have completely disappeared from the region in the 12 ka BP simulation during NH winter (Fig. 5c). Several lines of evidence from previous research have

suggested a southward shift of the ITCZ during the glacial, in particular during Heinrich events or the Younger Dryas (Atwood et al., 2020; Chabangborn et al., 2018; Fraser et al., 2014; McGee et al., 2014; Yang et al., 2020), and this is also simulated by CESM1 (Fig. 5). Main consequences of a southwardly displaced ITCZ are drier and cooler conditions on mainland Asia and consequently a drier and colder East Asian winter monsoon (Yang et al., 2020), and wetter conditions on the southern hemisphere (Ayliffe et al., 2013). In general, the ITCZ mainly responds to NH mean temperature which can be

altered on short time scales primarily by Heinrich events, or orbital forcing on longer time scales (Carolin et al., 2016, 2013; Deplazes et al., 2013; Otto-Bliesner and Brady, 2010). The simulated southward shift of the ITCZ and a complete absence of deep convection over ISEA during NH winter suggest that the ability of the ITCZ in causing rainfall in tropical Asia was greatly diminished at 12 ka BP over land areas. As atmospheric deep convection is turned off over land areas of the Sunda region, it is instead shifted east/west into the Pacific Ocean/Indian Ocean (Fig. 5). In winter, this implies a reversal of the

land-ocean circulation during 12 ka BP causing seasonal aridity. While proxies and models suggest a southward shifted ITCZ and drier and more seasonal conditions, a complete shutdown of deep convection and a dissolution of the ITCZ caused by the cold and dry East Asian winter monsoon during NH winter insolation minima, is a new insight from our study.

Figure 5 compares the January atmospheric circulation for 12 ka BP and PI. The large-scale wind pattern with strong

convergence over Sunda during the PI leads to deep convection and high precipitation amounts. However, the intrusion of cold and dry northeasterly winds during 12 ka BP, and relatively cold land temperatures over ISEA relative to warmer tropical SST, creates a highly divergent flow over Sunda at 12 ka BP, leading to subsidence and a breakdown of deep convection and seasonal dryness (Fig. 5b, 5d and Fig. 6). Figure 5b and 5d panels represent the vertical component of atmospheric circulation (Omega parameter; convective, $\omega < 0$ Pa/s) or sinking (stable, $\omega > 0$ Pa/s) air masses along a vertical

cross-section at the equator for 12 ka BP and PI that determine the atmospheric circulation cells over Indo-Pacific region. In addition, Fig. 6 displays areas of rising and sinking air masses for both periods on a wider spatial scale around the vertical cross-section in Fig. 5. Deep convection areas are represented by negative $\omega$ values extending beyond the 500 hPa level while local near-surface convection e.g., due to orography, may still occur on the 850 hPa level independent of the large-scale circulation. Both figures clearly show that deep convection over ISEA is drastically reduced or even inverted to into

subsidence at 12 ka BP, while convection and precipitation is shifted to the oceans. Near-surface (850 hPa pressure level) convection still takes place over Borneo and Java in January at 12 ka BP but does not extend further up to higher levels (≥500 hPa) in the atmosphere (Fig. 6). Considering convection (Fig. 5b, 5d and Fig. 6), surface winds (Fig. 5a, 5c) and zonal wind patterns (Fig. S7), we sketch the simplified Walker Circulation for both periods in panel 5b and 5d. With the results presented here, CESM1 suggests a breakdown of atmospheric deep convection, drastically reduced precipitation over ISEA

and complete re-arrangement of tropical atmospheric circulation over the Pacific and Indian Ocean at 12 ka BP compared to PI.

The shut-down of deep convection over ISEA and shift of deep convection to the surrounding oceans are closely connected to the reorganization of the Walker Circulation shown in Fig. 5. There is increased precipitation in the western Indian Ocean
at 12 ka BP, and a shift to more easterly surface winds (Fig. 5, Fig. S7). The Pacific trade winds were weaker at 12 ka BP; the mean annual wind speeds over the Pacific El Niño region 3.4 (5° N-5° S, 120-170° W) were 5.6 m/s on average, 12 % weaker than the 6.2 m/s annual average for PI. Figure S7 shows the mean zonal winds for 12 ka BP and PI in January, indicating a drastic slowdown or reversal of zonal winds over the oceans surrounding ISEA. The zonal SST gradient was also lower at 12 ka BP: the coastal waters west of South America were slightly warmer at 12 ka BP while the SST in the
IPWP was cooler (Fig. S3). An eastward shift of the convective arm of the Pacific Walker Circulation leads to El Niño–like climate conditions in terms of atmospheric circulation that result in a substantial decrease of rainfall in the ISEA region (Dang et al., 2020; De Deckker et al., 2003; DiNezio et al., 2016; Windler et al., 2019). The equatorial atmospheric circulation for January in our 12 ka BP simulation (Fig. 5d) closely reproduces the annual mean Indo-Pacific Walker Circulation in an LGM simulation using MRI-CGCM3 (Hollstein et al., 2018). The similarity is likely caused by cold NH
and exposed Sunda shelf leading to decreased ISEA convection in both time periods, but only has a significant impact on climate seasonality during the Lateglacial due to precessional forcing according to our TRACE seasonality results (Fig. 2 f,g).

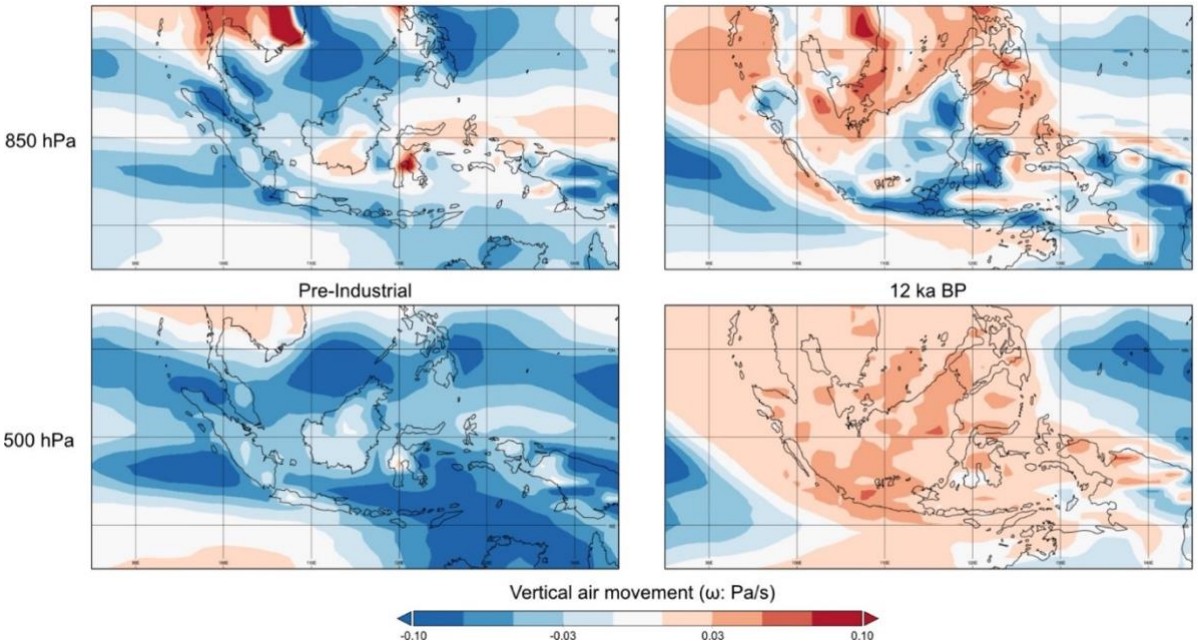

Figure 6. Comparison of near-surface convection/subsidence (850 hPa level) with (no) deep convection (≥500 hPa level) over ISEA during January for PI (left) and 12k (right). As in Fig. 5, negative values (blue) indicate rising air/convection and positive values (red) indicate subsidence/breakdown of convection. Note that near-surface convection due to orography around Borneo at 12 ka BP keeps all months humid (>60 mm) while the overall deep convection of the PI is completely shut down over the whole IPWP region causing at least seasonal aridity.


To put the mean changes of 12 ka BP relative to PI into context with modern ENSO climate states, we compare the monthly Walker Circulation Index (WCI) of the different periods with those to the modern (1979-2020) climate data from the ERA5 reanalysis dataset (Hersbach et al., 2020) in Fig. 7. The WCI analysis shows that while the Walker Circulation at 12 ka BP was similar in strength in NH summer compared to PI, it was drastically weaker and even reversed at 12 ka BP during NH

winters. In fact, the 12 ka BP winter Walker Circulation shows the same reversal as the composite average of the most extreme El Niño events of the reanalysis data (in 1983, 1998 and 2015) (Fig. 7). Moreover, the January mean state SST difference at 12 ka BP compared to PI qualitatively replicates the SST anomalies in El Niño events in that the eastern Pacific is anomalously warm compared to the western Pacific (Fig. 7b and 7c). The 12 ka BP climate was thus "El Niño-like" in terms of both atmospheric Walker Circulation and sea surface temperatures, and specifically so in NH winter. This suggests

that the 12 ka BP mean state resembles modern El Niño conditions in winters in terms of SST's and atmospheric circulation, and further supports previously hypothesized increased ENSO activity during the deglaciation based on proxy and model information (Clement et al., 1999; Koutavas and Joanides, 2012; Sadekov et al., 2013). It is important to note that our simulations use prescribed ocean states that are kept constant during each snapshot simulation, and we focus on the atmospheric responses on seasonal, monthly, and mean state changes over the entire 100-year snapshot. Due to the

prescribed ocean state, it is not meaningful to investigate the ENSO variability on interannual time scales caused by for example thermocline or SST variability (such as the Niño 3.4 index) in our CESM1 simulations. The TRACE winter Walker Circulation is weakened at 12 ka BP compared to PI, but TRACE very poorly reproduces the observational data and generally shows a too strong circulation and no reversal during 12 ka BP (not shown). This might be an indication that a reversed Walker Circulation in winter at 12 ka BP over ISEA is (at least partly) also due to a more realistic representation of

Sunda land exposure and not only due to a southward migration of the ITCZ which is found in both model simulations. Overall, it appears crucial to explore these changes at higher spatial model resolutions as key aspects are not represented by TRACE with ~415 km spatial resolution despite using the same ocean states as in CESM1 at ~120 km resolution.

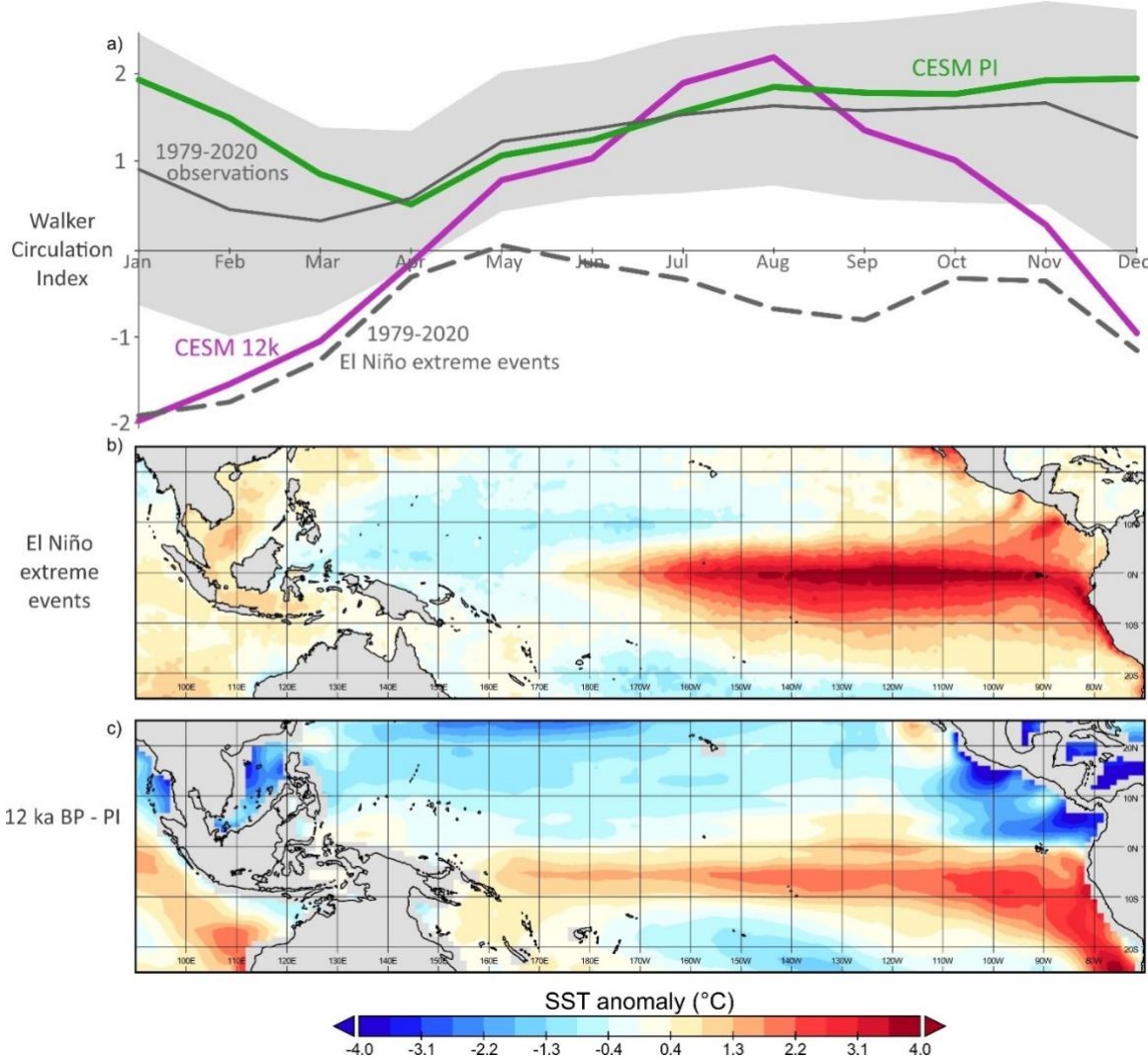


Figure 7. Comparison of the monthly mean Walker Circulation Index and SST anomalies for 12 ka BP and the modern states. a) Monthly mean Walker Circulation Index based on CESM1 simulations for PI (green) and 12 ka BP (purple) and ERA5 reanalysis data for 1979-2020 (gray) with shading denoting ±1 standard deviation in ERA5 monthly mean data. Also shown is the composite average over three most extreme El Niño extreme events (dashed gray line) from ERA5 (1983, 1998 and 2015). b) Average SST anomalies for the same three

El Niño events, and c) simulated average temperature difference at 12 ka BP compared to PI. The overall global cooling signal averaged over ±23.5° N around the equator has been subtracted in c). While the modern SST anomalies represent only extreme El Niño states, spatially comparable El Niño-like conditions dominate the mean climate at 12 ka BP over the tropical Pacific with highly negative Walker Circulation Index values in NH winter.

Increased precipitation in the western Indian Ocean combined with the low Sunda precipitation, and strengthened easterly winds over the Indian ocean (Fig. 5), resembles a positive Indian Ocean Dipole (IOD+) state, which has been shown to be caused by Sunda and Sahul shelf exposures during the LGM (DiNezio et al., 2016). This relationship between IOD+ and El Niño-like circulation is in agreement with the IOD dependence on ENSO (Stuecker et al., 2017). This is consistent with the model output for annual precipitation that shows a drier western Pacific and relatively wetter central Pacific (Fig. 5). We

therefore conclude that the mean climatic state that governed the tropical IPWP at 12 ka BP was largely modulated by a major reorganization of the Walker Circulation, resembling permanent El Niño and positive IOD states during NH winters in combination with a stronger East Asian winter monsoon and divergent flow over the IPWP region.

Putting our 12 ka BP results into a wider context, the broken up ITCZ and lower temperatures south of Java and northwest of
Australia, amongst the driest places in the simulation, may have been influenced by the lower sea level (DiNezio et al., 2016), but also by reduced heat transport from the Pacific to the Indian Ocean because of the reduced Indonesian Throughflow (ITF; Fig. 1) (Hendrizan et al., 2017). At 6.2 ka BP, the Indonesian Throughflow increased to modern levels in the TRACE simulation leading to a sudden winter wetting and hence decrease in seasonality (Fig. 2f). This sudden increase in through flow is an artefact of opening an ocean gateway at a low model resolution, and a more realistic representation of
ITF opening would be gradual over the early- to mid-Holocene sea level rise. Still, it is clear that the opened ITF water flow and heat flux cause drastically reduced seasonal differences in the region (Fig. 2) via increasing winter precipitation and reducing summer precipitation, as well as the general warming and decrease of insolation seasonality. According to TRACE, the sea level rise-induced opening of the ITF is thus the turning point in seasonality and transition to conditions resembling the present climate. Our CESM1 simulation agrees with earlier coupled climate model simulations of a closed Indonesian
Throughflow that suggested an El Niño-like climate mean state (Santoso et al., 2011; Song et al., 2007). An exposed Sunda shelf has earlier been suggested to have suppressed deep atmospheric convection in ISEA during the last glacial period (Bush and Fairbanks, 2003; Chabangborn and Wohlfarth, 2014). Consistent with our results of extreme seasonality, Thirumalai et al. (2019) suggest increased rainfall seasonality caused by the exposure of Sahul and Sunda shelves. Higher albedo of land compared to ocean when the Sunda shelf was exposed has also been proposed to cause lower convection and
moisture availability (DiNezio et al., 2016), which may explain part of the precipitation reduction seen in our simulation, since the Sunda shelf was still about 70 % exposed at 12 ka BP (Hanebuth, 2000; Voris and Sathiamurthy, 2006). Reduced heating of the exposed land would have significantly reduced the net energy input to the atmosphere (Byrne and Schneider, 2018) and thus reduced the intensity of the ITCZ (DiNezio et al., 2016). Alternatively, the aridity during the glacial has been attributed to the exposure of the Sunda shelf that shrunk the Indo-Pacific Warm Pool (Windler et al., 2019). Since the
seasonality simulated in TRACE follows insolation, which displays the opposite trend to the initial deglacial sea level rise, we suggest that insolation may have been a more important seasonality driver than sea level in Sunda during the Lateglacial, even though precessional cycles may play a smaller role than shelf exposure over longer glacial-interglacial time scales (Windler et al., 2019).

## 4.4 The relative importance of AMOC and other forcings during the Lateglacial

The major differences in Lateglacial forcing and boundary conditions compared to PI are more seasonal precessional forcing, increased freshwater input at high latitudes, lower greenhouse gas concentrations and remnant ice sheets with lower sea levels (Liu et al., 2014a). To disentangle the relative importance of freshwater forcing and resulting AMOC weakening

compared to other factors, we analyze a simulation of 13 ka BP (Allerød) when freshwater input was lower and AMOC was strong compared to 12 ka BP (Younger Dryas). The ocean state, reflecting the impact of meltwater forcing and changes in

AMOC, are prescribed from TRACE (He, 2011; Liu et al., 2014a) in our CESM1 simulations, as explained in the methods section. In TRACE, the meltwater input was 5 and 20 m/kyr for the Allerød and Younger Dryas simulations (He, 2011). As a result, the AMOC strength was decreased by 36 % from 14.5 to 9.2 Sv from the Allerød to Younger Dryas for the CESM1 snapshot simulations of 13 ka vs. 12 ka (Schenk et al., 2018). Additionally test how the Younger Dryas climate response to AMOC slowdown would have been without the changes in orbital forcing and boundary conditions using a sensitivity

simulation of 13 ka BP but with the ocean state of 12 ka BP, i.e., fully isolating the effect of reduced AMOC (13kYD simulation).

Comparing 12 and 13 ka BP relative to PI reveals that the drastically reduced ISEA precipitation and shift of convection centers occur during both periods (Fig. 8 a-d), and that the northern hemisphere is drastically colder, and with less cooling in

the tropics as discussed in section 4.2. The seasonal aridity is similar during both periods, suggesting that our findings for Younger Dryas applies to the longer term Lateglacial climate and is not only related AMOC slowdown in the Younger Dryas stadial. This is in line with the proxies described in section 3 since most proxies show a relatively small Younger Dryas signal relative to the Lateglacial into Holocene changes, and some records do not show any effect at all from the Younger Dryas (Krause et al., 2019; Partin et al., 2015). The difference between 12 and 13 ka BP over the (sub-)tropical climate zone

is almost exclusively forced by the AMOC shift. Changes in greenhouse gas concentrations, ice sheets and orbital forcing caused very little changes in precipitation, temperature, sea level pressure and winds according to our 13kYD hosing experiment (Fig. S8). This is different to high northern latitudes in summer where the increase in orbital forcing contributes to warmer summers in the Younger Dryas over the Eurasian continent (Schenk et al. 2018). The additional freshwater forcing in the Younger Dryas cools the North Atlantic by >5 °C, while the southern hemisphere warms slightly, (i.e., 'bipolar

seesaw' (Timmermann et al., 2010)), which results in an additional southward shift of the ITCZ at 12 ka BP relative to 13 ka BP (Fig. 8f). In terms of precipitation and seasonality in ISEA, the effects of the stadial AMOC shift are very small. However, there is apparent drying in the western equatorial Pacific, and wetting in central/east Pacific (Fig. 8f), suggesting a partly altered Walker Circulation, which occurs during June-October. Changes in AMOC (and ensuing cold high latitude northern hemisphere) are thus causing 1) a southward ITCZ shift over the Pacific Ocean and 2) rearrangement of the Walker

Circulation with drying in the West Pacific Warm Pool and wetting in the central/east Pacific. This finding supports our conclusions about mechanisms for changes in ISEA hydroclimate during the Lateglacial with an important role for a cold NH and large thermal meridional gradient described in section 4.3, but also suggests that short term AMOC changes can have an important additional effect on Walker Circulation changes. The change in Pacific Ocean precipitation patterns under reduced AMOC during the Lateglacial in our CESM1.0.5 simulations are consistent with climate model hosing simulations

using CESM1.2 in the pre-industrial climate state with reduced AMOC (Orihuela-Pinto et al., 2022). However, the atmospheric Walker Circulation changes for their 'AMOC-off' simulation under a pre-industrial state are opposite to our

results for Younger Dryas (i.e., they report La Niña-like conditions while our Younger Dryas results resemble El Niño conditions), confirming that the changes over Younger Dryas to PI time scale are mainly driven by other factors than AMOC.


The residual change in forcing between 12k and 13kYD (i.e., same ocean state but slightly different orbital, greenhouse gas concentrations and ice sheets) have very little impact (Fig. S8 e, f). The impact is mainly on the ISEA climate in autumn and spring, following the insolation change at each season; September has slightly increasing insolation from 13k to 12k, leading to slightly warmer and wetter autumn conditions at 12k, while the opposite is the case for March. Overall, this has a very

minor effect on the seasonality of ISEA climate.

From the results outlined above we conclude that AMOC shifts play a minor role for overall changes in the IPWP climate on long, glacial-interglacial time scales, i.e., Lateglacial relative to pre-industrial. Orbital, ice sheet and sea level forcings dominate the difference between our simulations of the pre-industrial and 12 or 13 ka BP, regardless of AMOC state in the

model. However, AMOC is a dominant forcing on shorter millennial (inter)stadial time scales, such as between 12 and 13 ka BP, where orbital (and residual) forcing causes almost no change as seen in the 13kYD sensitivity experiment. AMOC shifts can thus be an important modulator of both ITCZ and Pacific Walker Circulation on short time scales. Notably, little change is observed over ISEA and the Indian Ocean during the Lateglacial AMOC shifts in our simulations, which might be related to stronger teleconnections between AMOC shifts and the tropics via Atlantic-Pacific cross-basin interactions causing the

strongest impact in the Pacific Ocean (Yuan et al., 2018; Wang, 2019). However, we also note that Indian Ocean and ISEA proxies are sensitive to millennial scale AMOC events (Liu et al., 2022; Mohtadi et al., 2014).

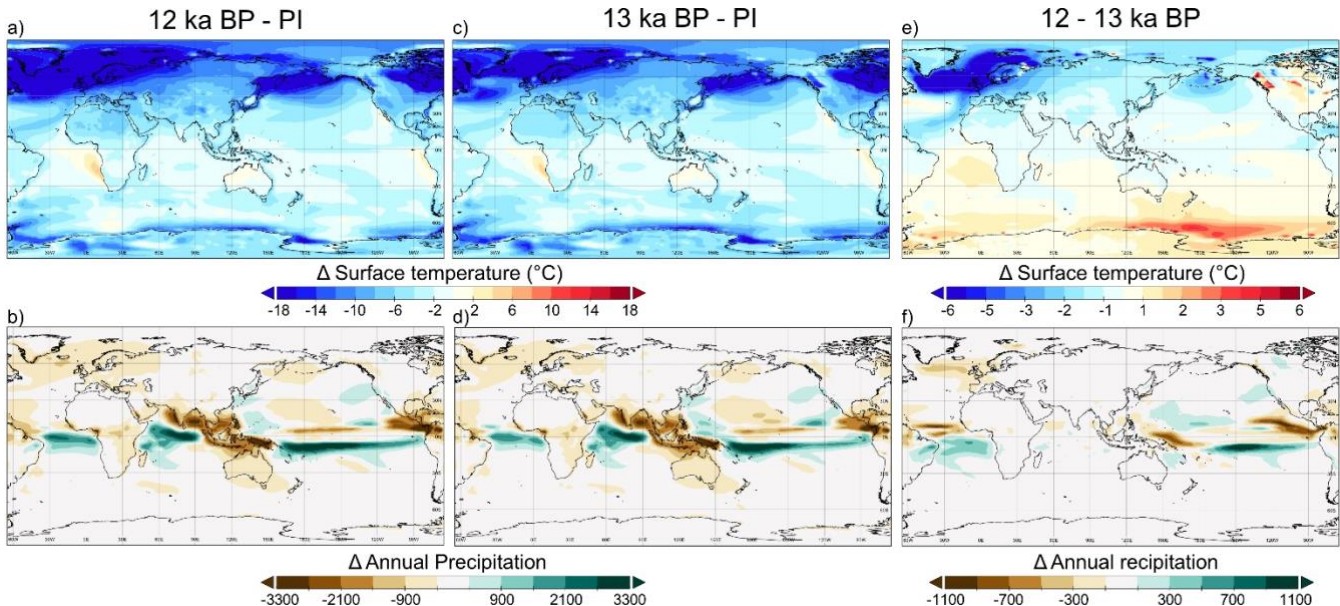

Figure 8. Comparison of pre-industrial, 12 and 13 ka BP climate states simulated by CESM1. The upper and lower rows show changes in temperature and precipitation, respectively. a-d) show that the changes between 12 and 13 ka BP relative to PI are very similar in the tropics. e and f) show changes between 12 and 13 ka BP which most notably consist of NH cooling, and SH warming, and that tropical Pacific precipitation patterns are shifted.

### 4.5 Implications for the "Savanna Hypothesis"

The simulated dry winters during the Lateglacial introduce a dry season in present-day ever-wet locations and increase the length of the dry season in other locations (Fig. 4). This must have caused a significant disturbance of the hydrological conditions of many ecosystems, in particular wetlands that could periodically be more prone to drying out. Together with mean annual precipitation of ~1500 mm in most coastal areas (Fig. 3), the Lateglacial climate may in general have favored a savanna biome (Lehmann et al., 2011). Indeed, proxy data from NE Thailand suggest that strong seasonality under glacial conditions in the Sunda shelf favoured the expansion of C4 grasses (Yamoah et al., 2021). Strong seasonality favours C4 grasses over C3 forest vegetation, because the former can cope much better with extended dry seasons due to a much better water use efficiency (Dubois et al., 2014; Yamoah et al., 2021). The only location that still receives rainfall throughout the year in our simulation (P > 60 mm/month, annual P > 2000 mm) is Borneo. Together with a minimum monthly surface temperature of 18-23 °C, this provides conditions favorable for ever-wet tropical rainforest vegetation, according to definitions by Köppen and biome reconstructions (Beck et al., 2018; Cannon et al., 2009). The simulated moderate reduction in precipitation on Borneo compared to the rest of Sunda is due to the orographic effect caused by the high mountains on the island, resulting in local near-surface convection and precipitation, despite the large-scale subsidence generally dominating ISEA (Fig. 5 and Fig. 6). Our results are hence also consistent with proxy records around Borneo (Chabangborn et al., 2018; Dubois et al., 2014; Schröder et al., 2018; Wang et al., 2007; Wurster et al., 2019, 2010) and peatland accumulation (Dommain et al., 2014). Dommain et al. (2014) report that the first wetlands to initiate peat accumulation (and are still remaining) in SE Asia were on inland Borneo at approximately 14-15 ka BP. Our simulations show that Borneo was the only location to not experience dry NH winters during 12 ka BP, giving a possible explanation as to why that was the first location for peat initiation.

Furthermore, our results suggest that seasonally dry glacial conditions may have lingered on the Sunda shelf until at least 12 ka BP, conditions that may have been suitable for a contiguous Savanna Corridor. While the environmental consequences to our modelling results are not simulated here, and need to be tested using vegetation modelling and further proxy research, the extreme shift in seasonality suggested by our simulations lend a robust modelling support for a Savanna Corridor to be present even during the Lateglacial. Based on the transient simulation of precipitation seasonality of TRACE (Fig. 2f), the combination of an opened Indonesian Throughflow, inundation of Sunda land and a decrease in seasonally diverging orbital forcing (Fig. 2e) may have successively turned ISEA into the ever-wet climate and ecosystem it is today during the early to mid-Holocene.

## 5 Conclusions

Using climate simulations for 12 and 13 ka BP and the pre-industrial climate state, a sensitivity experiment, a transient simulation from the Last Glacial Maximum to the present, and a proxy compilation from the region, we investigate how the mean climate and its seasonality has changed in Island South East Asia. We particularly focus on the identification of relevant changes and climate mechanisms preventing the establishment of tropical peats and rainforests under glacial and Lateglacial conditions. Our climate modelling results are in good agreement with proxy evidence for the generally drier annual conditions that prevailed at 12 and 13 ka BP and extended all the way back to the LGM. A more detailed analysis of the mechanisms behind the drier conditions highlights that changes during the NH winter climate were fundamentally different relative to today. Our simulations show that the (hydro)climate was much more seasonal, with the whole region except Borneo experiencing very dry conditions (<60 mm monthly precipitation) for several months per year. The spatial pattern of precipitation seasonality agrees well with climate proxies in the region. Notably, CESM1 correctly reproduces ever-wet conditions on Borneo due to local orography-induced convection that provides an explanation for why peat accumulation initiated there, while being impaired elsewhere in the region until the early-mid Holocene.

Key drivers for seasonally dry conditions resulting from differences between the Lateglacial and pre-industrial climates consist of 2 °C colder SST around ISEA – today's heat and moisture engine of the globe – and a ~27% larger thermal interhemispheric gradient under Lateglacial conditions between Siberia and Australia in NH winter, driving a stronger East Asian winter monsoon. Dry and cold northeasterly winds of the East Asian winter monsoon extended far south over today's IPWP region. As a result, winters were dry at 13 and 12 ka BP, with a complete collapse of tropical deep convection over ISEA due to diverging winds from land to sea and subsiding air masses over land areas. While lower precipitation due to a southward shift of the ITCZ during (late) glacial conditions has been proposed previously, CESM1 clearly shows that the reversal of the land-sea circulation over ISEA in NH winter acted as an independent mechanism dissolving any large-scale convergence of a potential ITCZ over land.

In summary, the main reason for the extreme precipitation seasonality changes relate to the suppressed deep convection over ISEA in response to a reduced winter insolation on the northern hemisphere, combined with Sunda shelf exposure that rendered land areas much colder than the surrounding oceans in NH winters. This, in turn, led to a major reorganization of the Walker circulation with a mean state in Lateglacial NH winters strongly resembling today's extreme El Niño events in the Pacific Ocean, and to a lesser extent positive Indian Ocean Dipole conditions. Using our results as analogues for future climate change in the IPWP, we suggest that the seasonal droughts simulated in the model and indicated by vegetation proxies under El Niño like mean state conditions in the Lateglacial, may provide clues to a future response of the earth system under global warming and projected changes in rainfall variability and ENSO. In particular, the paleo-record underlines the importance of closely tracking climate change with a focus on interhemispheric and seasonal divergence

which may introduce seasonal aridity in today's ever-wet ISEA region, with potentially severe consequences both for society and the tropical ecosystems and associated carbon stocks.

**Data availability**

The ERA5 data (Hersbach et al., 2019) can be downloaded from the Climate Data Store (CDS) (DOI:
10.24381/cds.f17050d7). CESM1 model data will be made available by Frederik Schenk (frederik.schenk@geo.su.se) upon reasonable request.

**Author contributions**

PH and FS designed the study in detail with broad guidance by RHS. CESM1 climate model simulations were conducted by FS. XK and PH analyzed the TRACE data. PH created the figures and proxy compilation. PH led the data analysis and PH
and FS wrote the manuscript with contributions from all authors.

**Competing interests**

The authors declare that they have no conflict of interest.

**Acknowledgements**

This research received funding from the Swedish Research Council to F.S. (VR 2015-04418) and R.H.S. (VR 2017-04430)
and Swedish Research Council for Sustainable Development to F.S. (FORMAS 2020-01000). X.K. received funding from the National Key Research and Development Program of China under Grant Nos. 2016YFA0600504 and 2017YFA0603803, and the National Natural Science Foundation of China under Grant No. 41775073. The climate model simulations with CESM1 were enabled by resources provided by the Swedish National Infrastructure for Computing (SNIC) at the National Supercomputer Centre (NSC), partially funded by the Swedish Research Council through grant agreement no. 2018-05973.
The CESM project is supported by the National Science Foundation and the Office of Science of the U.S. Department of Energy. TRACE was made possible by the DOE INCITE computing program, and supported by NCAR, the NSF P2C2 program, and the DOE Abrupt Change and EaSM programs.

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
