# Peer review of "El Niño-like Winter States Caused Seasonal Aridity in the Indo-Pacific Warm Pool During the Lateglacial"

_Climate of the Past, 2021_

## Author Response (AR1)

This document contains the point-by-point replies to the editor and reviewer comments, as well as the changes made to the manuscript during the review process.

*Grey italics are editor/reviewer comments.*

Black regular text is our first replies to the reviewers, outlining our suggested edits.

**Bold blue text is our point-by-point replies concerning how these edits were carried out.**

**Editor comments:**

*1) As mentioned by reviewer 1, the ability of the model to properly simulate precipitation over ISEA and ENSO variability should be mentioned in the main text, with a figure, similar to Fig_R1_1 included in the Appendix.*

**We have outlined a comparison between ERA5 precipitation and temperature to our present-day simulation in the main text at the end of the methods section (section 4.2, starting at line 183), and added a figure (Fig. S1) in the supplementary, as suggested.**

*2) I agree with reviewer 2 that the processes leading to the 12ka climatic anomalies have to be presented in a clearer manner. While I understand that a full decomposition of the forcing is probably out of the scope, separating the AMOC impact from the background conditions should be done. I thus agree with your proposed approach noting the following since there are little changes in boundary conditions (orb, ice-sheets, GHGs) between 12 and 13 ka.*
*- The AMOC strength in the forcing simulation (CCSM3) at 13 ka and 12ka should be clearly mentioned in the manuscript.*
*- The main ms should include maps of 12ka vs PI for the combination of AMOC and boundary condition changes, maps of 12ka vs 13 ka for the AMOC impact (13kYD -13k, and 12k-13kYD can be included in the Appendix). I would also suggest to include 13ka compared to PI in the main manuscript to provide an estimate of the impact of orb, ice-sheets and CO2, even though from Fig. R2_1 it is unclear if that would be very informative (if that's the case, then it could be moved to the Appendix).*

**We thank the editor for these suggestion and clarifications. We have added a clear mention and figure reference for CCSM3 freshwater forcing on line 505, and placed the figures in main manuscript and supplementary as suggested, and keeping 13ka BP – PI in the main manuscript to highlight the similarity between 13 and 12k relative to PI.**

*3) Typo: L. 131- GLAC1-D*
**Typo fixed.**

**Reviewer #1 comments:**

We sincerely thank the reviewer for taking the time and effort to read and comment on the manuscript. We think we can revise the manuscript according to the suggestions. We address each comment below, and have included the reviewer's comment in *grey italics* to aid the readability of this document.

*Reviewer #1: The article by Hällberg et al. presents an interesting and well thought study of seasonal variability of rainfall and climate over the last deglaciation in the Island South-East Asia (ISEA) region. The authors focus on how rainfall has changed regionally above the main landmass, that have themselves considerably been reshaped and shrinked along the deglacial eustatic sea level rise, and compare their transient model output with landmark paleoclimate (speleothem) reconstructions from different localities in the Indonesian archipelago. They detail, in particular, the seasonality which seems to have been greater at 12 ka as compared to nowadays, and manifest as the development of a dry austral summer dry period in the southernmost part of the ISEA. They discuss their results in light of peat deposits that started accumulating during the Holocene, leading to a potential savanna corridor connecting the main islands of Indonesia and plains in the Sunda and Sahul shelves prior to their exundation.*

*The analysis is nice, and I believe it should be published after some corrections that may be easy to deal with but could improve the manuscript, as detailed below.*

Replies to Reviewer #1

*Reviewer #1: First, the PI temp and precip patterns should be compared to modern data in Figure 3 or elsewhere, to highlight the ability of the model to reproduce local-scale climate variability, and even perhaps at seasonal and interannual timescales. The model seems to perform well, but rainfall seasonal and interannual variability at equatorial latitudes is, I think, notoriously challenging to reproduce with fully coupled GCMs. The authors should do better to convince the reader that the model is able to reproduce with a decent precision and accuracy at least the modern climate variability prior to inferring the transient evolution of past regional rainfall in such a complex environmental setting, with coastlines constantly varying.*

Reply: The CESM1 version used here is one of the scientifically validated releases by NCAR and is widely used in different studies including a validation of SE-Asian monsoon characteristics. Shown in Fig. R1_1 is the annual difference between the simulated and observed precipitation (presented at
[https://www.cesm.ucar.edu/experiments/cesm1.0/diagnostics/b40.1850.track1.1deg.006/atm_863-892-obs/](https://www.cesm.ucar.edu/experiments/cesm1.0/diagnostics/b40.1850.track1.1deg.006/atm_863-892-obs/) accessed 13-06-2022). These differences are also further discussed in the context of monsoons and ENSO by Meehl et al. (2020). The main discrepancies in the CESM1 simulations of modern climate compared to observations are 1) a westward displacement of precipitation maximum over the Indian Ocean, 2) the double ITCZ over the Pacific (see Zhang et al., 2019 for more details), and 3) a too strong Pacific Cold tongue (Meehl et al., 2020). The Maritime Continent precipitation is well reproduced by CESM1; small discrepancies consist of a slightly too dry Borneo, and slightly wet southern Indonesia. Since the model validation has been published previously in detail, we do not see a benefit of including a comparison in this manuscript. The remaining question is how realistic the response to late glacial conditions simulated in our 12k and 13k runs are, and the only way to validate the paleoclimate runs is to compare with proxy evidence. We show from the proxy compilation in our manuscript that the CESM1 in general is consistent with proxies from 12 ka BP.

We propose to refer to the CESM1 diagnostics webpage and the Meehl's paper in the methods section of our manuscript, to refer the reader to CESM comparisons previously done by others.

References:
Zhang, G. J., Song, X., Wang, Y. (2019) The double ITCZ syndrome in GCMs: A coupled feedback problem among convection, clouds, atmospheric and ocean circulations. Atmospheric Research, 229, 255-268, https://doi.org/10.1016/j.atmosres.2019.06.023.

Meehl, G. A., Shields, C., Arblaster, J. M., Annamalai, H., & Neale, R. (2020). Intraseasonal, seasonal, and interannual characteristics of regional monsoon simulations in CESM2. Journal of Advances in Modeling Earth Systems, 12, e2019MS001962. https:// doi.org/10.1029/2019MS001962

[Figure]

Figure R1_1: Precipitation bias in CESM1 for the modern climate. Sourced from the diagnostics of CESM1 (sourced from https://www.cesm.ucar.edu/experiments/cesm1.0/diagnostics/b40.1850.track1.1deg.006/atm_863-892-obs/ , accessed 26-04-2022).

**As indicated in the response to the editor, we added a section at the end of the methods (section 4.2, starting at line 183), and included a figure in the supplementary material, outlining model performance and biases.**

*Reviewer #1: I think you may also want to discuss in greater details the large-scale outputs of PMIP4 outputs to highlight such point: those recent results indicate extreme variability in LGM rainfall model output scattering in the ISEA region (Kageyama et al., 2021, their figure 6 is particularly instructive).*

Reply: We're well aware of these quite mixed results for the LGM. The large disagreement even on the sign of precipitation changes between models is rather worrisome although there is a somewhat better agreement that at least parts of the ISEA region were definitely drier in most simulations. Highlighting the spatial complexity and disagreement between models for the glacial period in this region is a good suggestion, and we would like to add this in the introduction in line 70, where we already discuss some diverging results in model-model and model-proxies. The high-resolution simulation figures in our study clearly show large spatial changes on short distances between very wet and very dry regions – which could explain the large disagreement in the PMIP4 results shown by Kageyama et al. 2021.

**We added a discussion about this in the model validation section (section 4.2, starting at line 183).**

*Reviewer #1: I also think that some important data available should be cited and discussed as follows:*
*- On ENSO, you cite Clement, Koutavas and Sadekov to suggest that ENSO was enhanced during the LGM, without discussing the contrasting results of Leduc et al., 2009, Ford et al., 2015 and Liu et al., 2017 that point to a reduced ENSO variability during the LGM. Those results that demonstrate that the LGM ENSO state is still debated should be explicitly discussed along with the Clement, Koutavas and Sadekov articles.*

Reply: We agree that this section needs to be revised and will rewrite it to better reflect the current state of the debate.

**This paragraph is now rewritten starting at line 52:**
**"There is also much debate about the past variability and states of ENSO. Both modelling and proxy evidence indicate weakened ENSO variability and "La Niña-like" conditions with a strong zonal sea surface temperature (SST) gradient during the mid-Holocene, prior to an increase in ENSO variability towards the near-present state in the Late Holocene (Brown et al., 2020; Carré et al., 2021; Chen et al., 2016; Emile-Geay et al., 2016; Grothe et al., 2020). However, the state of ENSO is not as well constrained during the Lateglacial (~14.7-11.7 ka BP) and the LGM (~21 ka BP). There is evidence for increased ENSO variability during both the LGM and Lateglacial, under El-Niño like states in terms of weaker Pacific SST gradients and larger δ18O variability in foraminifera (Clement et al., 1999; Koutavas and Joanides, 2012; Sadekov et al., 2013), but these interpretations have been challenged. Instead, a strengthened seasonal (annual) cycle (Ford et al., 2015; Zhu et al., 2017) and a supressed interannual ENSO variability has been proposed, being consistent with both proxies and model simulations (Ford et al., 2015; Leduc et al., 2009; Liu et al., 2014a; Zhu et al., 2017). A recent PMIP4 climate model ensemble indicates weakened ENSO variability at the LGM, but there is a large spread between different models (Brown et al., 2020)."**

*Reviewer #1: -I wonder why you choose to not discuss leaf wax isotopes (the Niedermeier and Konecky papers), other speleothem records (Griffiths), etc. without attempting to highlight them in Figure 2, as you've done for the tree other speleothem records. Also, the original paper evidencing a dry and open vegetation during the LGM at the Konecky site is Russel et al., 2014, and should also be cited, as dD interpretation is puzzling at this site as evidenced in Konecky.*

Reply: Niedermeier et al's paper on leaf wax isotopes are discussed in line 197 in the manuscript, stating that they report large variability, but no clear trends since the LGM to the present, which they interpret to be caused by local effects. Since the nearby speleothem d$^{18}$O record (Wurtzel et al., 2018) show a much more consistent trend to other maritime continent records and large-scale convection, we put more emphasis on Wurtzel's results. The Konecky et al. paper is included in the overview figure, but not further discussed in the text. Commenting or re-interpreting (and therefore showing) all these proxy records individually with respect to potential seasonality effects would be rather tricky, as these are also influenced by 'source' effects and 'amount' effects, and distract from our main message. Since the three speleothem records in ISEA that we show in Fig. 1 give a relatively good overview of similarities (and differences such as from southward shifts of ITCZ during the NH cold events) on a south-north transect across the equator, we think that more records are not generally useful to be added to this figure.

The Griffiths et al., (2009) speleothem d$^{18}$O record is also included in Ayliffe's (2013) composite which we refer to, but we agree that it is probably appropriate to also refer to Griffiths' original publication. Other records that we have left out (such as Partin et al., 2015 Palawan speleothem) only covers a shorter time period and does not allow comparison between 12ka BP and the near-present.

We will add the Russel et al., 2014 reference for the Tuwoti vegetation findings.

**We added the Griffiths et al. (2009) and Russell et al. (2014) references in relevant places. We agree that the Konecky discussion about difficulties in reconstructing precipitation isotopes in Sulawesi is very interesting, but is mainly of relevance for their LGM results, and thus not important in terms of the period around 12 ka BP and the scope of our manuscript.**

*Reviewer #1: -I also think you should better discuss what has happened at 12 ka in more details in the data as long as you opt for focusing on that time period. The results obtained in the simulation does not always fit to the data you highlight in Figure 2 for that time period, and there is a kind of overshoot in precip seasonality after 12 ka, when then YD resumed. I understand you can't describe everything in its full complexity but your statements for the 12 ka in particular are not always met in reconstructions, and a lot happens before and after 12 ka in both data and climate simulation, as shown in Figure 2.*

Reply: Since the focus of our manuscript is the difference between YD and the modern climate, we have decided to mainly discuss the large-scale differences between those periods, and not the (also interesting) details of climate responses to northern hemispheric short term forcings of the YD event (such as 12k vs 13k). We will expand on this in the revised version (see below).

We already explain some of the differences between the records for YD (ITCZ shift south, causing wetter Java signal and drier conditions to the north). The overshoot in climate seasonality after YD according to TRACE is primarily related to an increase in summer precipitation when the early Holocene warming commenced, and winters remained cold and dry. The seasonal distribution of precipitation clearly follows the precessional cycle where the summer-winter difference peaked at ~11 ka BP, with the lowest winter insolation on the northern hemisphere. We propose to add this reasoning to the manuscript. The seasonality is then rapidly diminished to near modern values upon the opening of the ITF at 6.2 ka BP in the model, and as discussed in the manuscript, this reduction in seasonality would have been more gradual during at least the Early Holocene as the ITF opened in response to sea level rise.

Since we will add a comparison with a 13k simulation in response to reviewer 2's suggestion, we also agree that further discussion about the changes in climate before and after the YD is appropriate to add, in light of those results.

**We added the reasoning outlined above in the manuscript section 4.1, from line 270.**

*Reviewer #1: Some other minor details:*
*-Fig 5b & d, I don't really see stronger vs weaker trade winds in the lower branch of the Hadley cell in panels a and c, but rather an eastward displacement of the convection site above the western Pacific at 12 ka WRT nowadays*

Reply: We agree that shifted convection centres is a main feature of the altered atmospheric circulation, and we write this in multiple places in the manuscript. We also agree that the weakening of trade winds is not easily seen from the wind vectors shown in figure 5a and c. However, in the text, on line 374 we state that the Nino3.4 zonal winds are 12% weaker. We also discuss zonal wind changes and in fig S6c where we show the zonal wind speed change between 12k and PI which are clearly weakened over the Pacific. We also refer to the zonal winds weakening on line 338, where we discuss figure 5.

To remedy that this may not be entirely clear to the reader, we propose to add the following sentence to the Figure 5 figure text: "For details on the weakening and/or reversal of the zonal winds, see figure S6." Fig. 5 is intended as a somewhat simplified sketch focusing on shifts in convection cells and adding horizontal wind anomalies would make the figure less readable.

**We added "For details on the weakening and/or reversal of the zonal winds, see figure S7." in the figure text to Fig. 5.**

*Reviewer #1: -Still on Figure 5, the situation shown in panel c looks quite like the LGM situation as seen in Holstein et al., 2018 (their figure S6), it is interesting to note that the LGM and YD have the same effect on the convective cell displacement during those two contrasting yet cold time periods*

We agree it is interesting that the LGM simulation using MRI-CGCM3 and our 12 ka BP CESM1 simulation show very similar effects on the Walker Circulation, despite different boundary conditions. We ascribe the reversed land-ocean circulation mainly to cooling of South East Asia during lower sea level and cold NH winters due to insolation and ice sheets. Since the sea level was even lower and the NH was very cold at the LGM, it is consistent with our findings that the LGM circulation may have been similar in some respects, despite that orbital forcing was very different. However, according to the TRACE results we present in figure 2, the LGM cooling of the Maritime Continent did not result in a more seasonal climate. We would like to add a brief discussion of this in the discussion section of our manuscript.

**We added this discussion from line 426.**

*Reviewer #1: -You state in the discussion and the conclusion that the deglacial interval « resembles to ENSO », but you show only variability and seasonal features. Is it possible to differentiate variability in rainfall that occurred at interannual timescale? The article by Liu et al., 2014, may help.*

Reply: Since our CESM simulations were run with a prescribed ocean state from TRACE, i.e., doesn't change during each snapshot simulation, the simulated variability in precipitation or ENSO in response to SSTs will be absent compared to a fully coupled simulation, and not meaningful to discuss in the case of our simulations. We realize that the wording is unclear where we write about resemblance to "El-Niño like" states. We propose to correct this by being more specific in stating that the results resemble El-Niño like mean states "in terms of sea surface temperature and sea level pressure patterns, and atmospheric circulation", and to be clearer that our results are concerning the seasonal cycle and mean state, not the short term (~2-7 year or similar) interannual ENSO variability, throughout the manuscript.

**We added this clarification to the discussion section where we discuss ENSO:**

**Line 26, more specific: "The altered atmospheric circulation, sea surface temperature and sea level pressure patterns during winters led to conditions resembling extreme El Niño events in the modern climate and a dissolution of the Inter-Tropical Convergence Zone (ITCZ) over the region"**

**Line 56, more specific: "There is evidence for increased ENSO variability during both the LGM and Lateglacial, under El-Niño like states in terms of weaker Pacific SST gradients and larger d$^{18}$O variability in foraminifera"**

**Line 422, a clarification that the El-Niño like state refers to atmospheric circulation.**

**Line 437: "The 12 ka BP climate was thus "El Niño-like" in terms of both atmospheric Walker Circulation and sea surface temperatures."**

**Line 441: "It is important to note that our simulations use prescribed ocean states that are kept constant during each snapshot simulation, and we focus on the atmospheric responses on seasonal, monthly, and mean state changes over the entire 100-year snapshot. Due to the prescribed ocean state, it is not meaningful to investigate the ENSO variability on interannual time scales caused by for example thermocline or SST variability (such as the Nino 3.4 index) in our CESM1 simulations."**

*References:*
*Ford, H. L., Ravelo, A. C., & Polissar, P. J. (2015). Reduced El Niño–Southern oscillation during the last glacial maximum. Science, 347(6219), 255-258.*
*Hollstein, M., Mohtadi, M., Rosenthal, Y., Prange, M., Oppo, D. W., Méndez, G. M., ... & Hebbeln, D. (2018). Variations in Western Pacific Warm Pool surface and thermocline conditions over the past 110,000 years: Forcing mechanisms and implications for the glacial Walker circulation. Quaternary Science Reviews, 201, 429-445.*
*Kageyama, M., Harrison, S. P., Kapsch, M. L., Lofverstrom, M., Lora, J. M., Mikolajewicz, U., ... & Zhu, J. (2021). The PMIP4 Last Glacial Maximum experiments: preliminary results and comparison with the PMIP3 simulations. Climate of the Past, 17(3), 1065-1089.*

*Leduc, G., Vidal, L., Cartapanis, O., & Bard, E. (2009). Modes of eastern equatorial Pacific thermocline variability: Implications for ENSO dynamics over the last glacial period. Paleoceanography, 24(3).*

*Liu, Z., Lu, Z., Wen, X., Otto-Bliesner, B. L., Timmermann, A., & Cobb, K. M. (2014). Evolution and forcing mechanisms of El Niño over the past 21,000 years. Nature, 515(7528), 550-553.*

*Russell, J. M., Vogel, H., Konecky, B. L., Bijaksana, S., Huang, Y., Melles, M., ... & King, J. W. (2014). Glacial forcing of central Indonesian hydroclimate since 60,000 y BP. Proceedings of the National Academy of Sciences, 111(14), 5100-5105.*

*Zhu, J., Liu, Z., Brady, E., Otto Bliesner, B., Zhang, J., Noone, D., ... & Tabor, C. (2017). Reduced ENSO variability at the LGM revealed by an isotope enabled Earth system model. Geophysical Research Letters, 44(13), 6984-6992.*

Reply: We will add these references suggested above by reviewer #1 to the manuscript, as indicated in the replies above. **These references have been added.**

**Reviewer #2 comments:**

We thank reviewer 2 for the very useful comments and suggestions. As addressed further below, a key point raised by the reviewer is the need to compare our results presented in the manuscript with an additional simulation to isolate the effect of different forcings. We agree and believe that the addition is useful for the manuscript, and we outline a potential revision below, in which we add additional model results from the 13k simulation data, as well as another sensitivity experiment of 13k with an ocean state from 12k, i.e., increased freshwater forcing. This sensitivity experiment allows us to isolate the effect of AMOC-only changes vs. orbital forcing changes of 13k vs. 12k. We address each comment below, and have included the reviewer's comment in *grey italics* to aid the readability of this document.

*Reviewer #2: Overall, I like the concept of the manuscript: describing seasonal changes in western tropical precipitation during the Younger Dryas climate event. However, I cannot accept the manuscript in its current form, and I suggest major revisions to the manuscript. The primary cause for the revision is to clearly separate which forcings are responsible for differences between the Younger Dryas state and Pre-Industrial (PI).*

*Reviewer #2: In the intro (paragraph on lines 49-55): the jury is still out as to how ENSO changed during the glacial times, including during the deglaciation. This paragraph and the paper should depict the current state of ENSO research. Reviewer #1 provides many important references.*

Reply: As we responded to reviewer 1, we agree that this section needs to be revised, and will rewrite it to better reflect the current state of the debate.

**This paragraph is now rewritten starting at line 52:**
**"There is also much debate about the past variability and states of ENSO. Both modelling and proxy evidence indicate weakened ENSO variability and "La Niña-like" conditions with a strong zonal sea surface temperature (SST) gradient during the mid-Holocene, prior to an increase in ENSO variability towards the near-present state in the Late Holocene (Brown et al., 2020; Carré et al., 2021; Chen et al., 2016; Emile-Geay et al., 2016; Grothe et al., 2020). However, the state of ENSO is not as well constrained during the Lateglacial (~14.7-11.7 ka BP) and the LGM (~21 ka BP). There is evidence for increased ENSO variability during both the LGM and Lateglacial, under El-Niño like states in terms of weaker Pacific SST gradients and larger ▢18O variability in foraminifera (Clement et al., 1999; Koutavas and Joanides, 2012; Sadekov et al., 2013), but these interpretations have been challenged. Instead, a strengthened seasonal (annual) cycle (Ford et al., 2015; Zhu et al., 2017) and a supressed interannual ENSO variability has been proposed, being consistent with both proxies and model simulations (Ford et al., 2015; Leduc et al., 2009; Liu et al., 2014a; Zhu et al., 2017). A recent PMIP4 climate model ensemble indicates weakened ENSO variability at the LGM, but there is a large spread between different models (Brown et al., 2020)."**

*Reviewer #2: The major issue: by comparing 12 ka to PI, the authors are conflating several different forcings that could drive the precipitation changes, such that it is not clear if the conclusions hold as presented. The summary of the difference in forcings between 12 ka (Younger Dryas) and 0 ka are:*
*Precession of the equinoxes.*
*Sea Level changes and the presence of large ice sheets on land*
*Freshwater forcing reducing AMOC*

*To separate the forcings, and hence attribute what is causing the differences between the paleo and PI runs, the authors should compare different time slices, in addition to the 12 ka and 0 ka.*
*Usually the early Holocene (~10 ka) or the mid-Holocene (~6 ka) compared to 0 ka is used to identify changes forced by precession of the equinoxes, like in the PMIP protocol. This is \*especially important when considering changes in the seasonal cycle, as precession has little effect on mean annual changes in insolation, but instead causes very large impacts on the seasonal distribution of insolation (e.g. Huybers, Science, 2006 or Huybers, QSR, 2007). The insolation changes in turn lead to changes in precipitation. Therefore, the authors need to make major changes to the manuscript, as the seasonal cycle is a major point of focus for this manuscript.*

*Comparing the time interval just before the Younger Dryas, i.e. the Bølling–Allerød interstadial (BA) at 13-14 ka, with 0 ka will address both orbital changes and sea level/ice sheets. By then using the 10 ka run, which addresses precession only, and the ~13 ka run, one can isolate the climate effects due to sea level and ice sheets.*

*Comparing the run during the BA with the YD will isolate the climate effects due to AMOC changes.*

*I am not suggesting to do any additional runs, such as 10 ka. Rather, present what runs have been completed, and present the output accordingly. Schenk, et al., Nature Comm., 2018, ran the BA interval. So that should be readily available to the*

*authors. The 0, 12, and 13 ka runs should be used to better describe which forcings are driving the seasonal changes in precipitation of the western tropical Pacific.*

Reply: The main focus of this manuscript was to explore the differences in climate between the Holocene and deglacial periods, in context of absence of peat formation and more open vegetation during the glacial. As such, it is outside the scope of this paper to fully isolate which changes in forcing are responsible for the large scale deglacial-Holocene transition. With that said, we agree with the reviewer that the comparison with 13 ka BP simulation would be useful as an addition to the analyses already presented, even though we will not be able to fully isolate all forcings and changes in boundary conditions. We thus propose to add a section to the manuscript with the comparison suggested by the reviewer for 13k vs. 12k, with the figures and main points as discussed below.

Additionally, we have access to an additional simulation experiment with 13 ka BP climate state, but with the cold ocean state from 12 ka BP (which we refer to as 13kYD). This allows to isolate the changes in climate between 13k and 12k that came only from ocean SST's (i.e., AMOC/meltwater) changes, and what changes arose from other forcing (orbital, GHG, sea level). The comparisons we propose in this new section are thus:
1) 12k and 13k versus PI: large scale changes deglacial to modern forced by GHGs, orbital, ice sheets and sea level. Essentially here we compare deglacial to modern conditions, as already done in the manuscript but with one more simulation.
2) 12k vs 13k: full changes between YD and Allerød
3) 13k vs 13kYD: isolates AMOC change
4) 12k vs 13kYD: isolates the residual of changes arising from non-AMOC related forcing, i.e. mainly orbital + nonlinearities

Results 13k simulations:
1)
The new comparison shows that the main differences between 12k-PI are the same as 13k-PI (fig RC2_1 a-d), with drastically reduced ISEA precipitation and shift of convection centers during both periods. That the difference between 12k and 13k is relatively small compared to PI suggests that our original approach is valid for our goal to compare late glacial and PI; it does not matter much which specific time period during the late glacial is used if the focus is on a comparison with PI. This was also the reason why we did not include 13k in the current manuscript (results for 13k also show a prohibitive climate for peatland formation as for 12k).

2)
As the reviewer suggests, the 12k-13k comparison is informative, and we see changes in Pacific circulation and seasonal cycle that are related both to the changes in AMOC and precession (Fig RC2_1 e and f). The increased freshwater forcing at 12k cools the north Atlantic and to a lesser extent the entire northern hemisphere.

The (annually averaged) precipitation centers are shifted further south caused by the southward shift of ITCZ during NH cooling – which is a well-known phenomenon as discussed in the manuscript on line 357. There are only minor changes over ISEA land areas between the periods. There is apparent drying in the western equatorial Pacific, and wetting in central/east Pacific, suggesting a partly altered Walker Circulation. In terms of seasonality of this simulated annual change in precipitation between 12k and 13k, there is little effect over ISEA, but the substantial drying in the western equatorial Pacific occurs during June-October, while the November-May period mainly shows a southward shifted ITCZ.

[Figure]

Figure R2_1. The change in TS and P between 12k, 13k and PI. a and b) 12k-PI, c and d) 13k-PI, e and f) 12k-13k.

3)
The differences between 13k and the 13kYD simulations reveal that almost all changes between Allerød and Younger Dryas were caused by AMOC changes, since nearly the whole difference between 13k and 12k (Fig. RC2_2 a and b) is also captured in 13k-13kYD (Fig. RC2_1 c and d). Changes in AMOC (and ensuing cold high latitude northern hemisphere) are thus causing a 1) southward ITCZ shift over the Pacific Ocean and 2) rearrangement of the Walker Circulation with drying in the West Pacific Warm Pool and wetting in the central/east Pacific. This finding supports the conclusions made in the original manuscript, concerning the role of a cold NH and large thermal meridional gradient, but also suggests that short term AMOC changes can have an important effect on Walker Circulation changes.

4) The residual difference between 12k and 13kYD (i.e., same ocean state but slightly different orbital, GHG and ice sheets) show very small differences (Fig. RC2_2 e and f). These differences appear to mainly affect the ISEA climate in winter and spring, following the insolation change at each season. September has slightly increasing insolation from 13k to 12k, leading to slightly warmer and wetter autumn conditions at 12k, while the opposite is the case for march. Overall, this has a very minor effect on the seasonality of ISEA climate.

Conclusions:
From the results we outlined above we conclude that AMOC is the dominant forcing on short (millennial) time scales, such as between 12 and 13 ka BP. AMOC can thus be an important modulator of both ITCZ and Pacific Walker Circulation. Little change is observed over ISEA and Indian Ocean due to AMOC. On short (millennial) time scales, orbital forcing plays a minor role. On longer timescales between 12 or 13 ka and PI, AMOC changes (as reflected by model freshwater forcing and resulting ocean SSTs) play a minor role. On these longer time scales, the orbital, ice sheet and sea level forcings dominate.

[Figure]

Figure R2_2. Precipitation and surface temperature changes between the 12k and 13k simulations, and the sensitivity experiment. a and b) 12-13 ka BP, representing the total change between the simulations. c and d) 13kYD-13 ka BP,

representing AMOC only. And e and f) 12 ka BP-13kYD, showing the residual changes, arising from small changes in GHG, orbital, sea level. Also shown are the anomalies in surface winds as vectors in a, c and e and tropical (±23.5°N) sea level pressure difference as isobars in b, d and f where dotted isobars denote negative values and solid isobars denote positive values. Note that the strong temperature changes in Sunda and Sahul shelves are artefacts arising from comparing land to ocean grid points resulting from the altered sea level between simulations.

**We added a new section to the manuscript (Section 4.4, starting at line 499) outlining these results, and added relevant information to the method and abstract sections.**

*Reviewer #2: Plot the coast lines in Figure 3-7 using the boundary conditions in the model, not modern observations of coastlines or bathymetry. This greatly facilitates the interpretation of the precipitation signals. One should look at where the model has land – not the actual planet.*

Reply: The model land-sea distribution for 12ka BP in the model was shown in figure 7c in the original manuscript, which is pointed out in the methods. We left the model coastlines out of figure 3-6 to improve readability, and not make the figures excessively busy, in particular for figures with several panels combined. The bathymetric coastlines that we currently use in the figures also reflect the model coastline relatively well. We thus suggest adding the coastlines to figure 3 only, which both allows for better readability while still clearly showing the coastlines for both PI and YD to the reader. However, it is relatively easy to plot the model coastlines in all figures if the editor or reviewer disagree with this suggestion.

**We have added the coastlines to figure 3, and updated the method's section and figure 3 figure text accordingly. Since this renders the figures harder to read and is very similar to bathymetric coastlines already used, we would like to refrain from applying this to the rest of the figures of the manuscript.**

*Reviewer #2: If you want to use the term "El Niño like" to describe the mean state, one should include changes in the thermocline and sea surface salinity (Di Nezio, Paleo., 2011). Sea surface temperatures alone are not sufficient to describe the mean state when discussing changes in ENSO. Furthermore, what is the ENSO signal in the runs? 150 years is not adequate enough to converge on the median of the ENSO response (Lawman, et al., Science Adv., 2022), but it would be nice to see how the Niño 3.4 SSTA at 12 and 13 ka responds in CESM1, as it has a realistic ENSO.*

Reply: Since this simulation was done with a prescribed ocean state from TRACE, i.e., is fixed for the duration of the snapshot simulations, the short term (2-7 year or similar) ENSO variability in not meaningful to discuss in this paper. We therefore focus on changes in the mean state and seasonal cycle - and use the term "El Niño like" to describe the patterns in SST, sea level pressure, winds and large-scale atmospheric circulation. We agree however that this should be clarified in the manuscript text better, and will be more specific in what we mean when we write "El Niño-like". We acknowledge that DiNezio's suggestion to use SSS and thermocline would be useful in a fully coupled simulation, but in our case that does not make sense since the ocean does not respond dynamically to changes in the atmosphere.

We thus believe that the term "El Niño-like" in relation to SST patterns, SLP and atmospheric circulation is a valid and useful term to describe the mean state simulated by the model, as long as we specify clearly what we mean. We note that this is an established usage of the term, and seen in other recent papers (such as Brown et al., 2020, and references therein).

Brown et al., 2020 reference: Climate of the Past, 16, 1777–1805, https://doi.org/10.5194/cp-16-1777-2020

**We updated the manuscript in several places to be more specific in what we mean by 'El Niño-like', as we wrote in response to reviewer 1:**

**Line 26, more specific: "The altered atmospheric circulation, sea surface temperature and sea level pressure patterns during winters led to conditions resembling extreme El Niño events in the modern climate and a dissolution of the Inter-Tropical Convergence Zone (ITCZ) over the region"**

**Line 56, more specific: "There is evidence for increased ENSO variability during both the LGM and Lateglacial, under El-Niño like states in terms of weaker Pacific SST gradients and larger d$^{18}$O variability in foraminifera"**

**Line 422, a clarification that the El-Niño like state refers to atmospheric circulation.**

**Line 437: "The 12 ka BP climate was thus "El Niño-like" in terms of both atmospheric Walker Circulation and sea surface temperatures."**

**Line 441: "It is important to note that our simulations use prescribed ocean states that are kept constant during each snapshot simulation, and we focus on the atmospheric responses on seasonal, monthly, and mean state changes over the entire 100-year snapshot. Due to the prescribed ocean state, it is not meaningful to investigate the ENSO variability**

**on interannual time scales caused by for example thermocline or SST variability (such as the Nino 3.4 index) in our CESM1 simulations."**

*Reviewer #2: Please update the Borneo record to use the recently published version (Buckingham, et al., GRL, 2022). It has a clear YD signal. I realize that this manuscript was submitted before the GRL manuscript was published. This is not to penalize, but rather to update, this manuscript. Also, the interpretations in that manuscript may help strengthen this manuscript.*

Reply: We thank the reviewer for this suggestion that we indeed had not published at time of submission. We will update the manuscript accordingly.

**We added Buckingham et al. (2022) speleothem record to figure 2.**

Other edits:

- To make the manuscript title clearer, we changed it from "El Niño-like conditions and seasonal aridity in the Indo-Pacific Warm Pool during the Younger Dryas" to " El Niño-like Winter States Caused Seasonal Aridity in the Indo-Pacific Warm Pool During the Lateglacial ". Since we included an extra simulation of 13k in addition to 12k, it is now more appropriate to use "Lateglacial" than Younger Dryas. We also clarify that the seasonality is caused by the winter state.
- The abstract was updated slightly to incorporate the new results from analysing the 13 ka BP simulation.
- Added a clarification in section 2.3 that the WCI calculation unit is hPa.
- Several language and spelling edits to improve readability.
- Minor spelling edits and changes in units and legend labels for consistency (such as always using °C for temperature, instead of K for temperature changes).
- Figure S4 and S7 color bar had slight errors which are now corrected.
- Since we added a simulation of Allerød, it is no longer suitable to refer to Younger Dryas in many places, we instead use 'Lateglacial' which covers both.
- Using 'Younger Dryas' instead of YD acronyms, to improve readability.
- Added the word "reversal" in an additional place in Fig. 5d for highlighting purposes.

---

## Author Response (AR2)

- Added the values in Sv for AMOC strength in Allerød and YD at line 526.
- About using "winter" and "summer": we have now specified explicitly that we refer to northern hemisphere (NH) seasons, in the beginning of the manuscript, and have made this also clearer in the text by adding 'NH' at many places. In this way, 'NH winter' refers to that particular time of the year without specifying the exact months, instead of referring to an actual cold season, which indeed is not the case in the tropics. Using the DJF and JJA is not always suitable in the paper, since we show figures of only January.
- We adjusted the title in line with not using 'winter' to: "Seasonal Aridity in the Indo-Pacific Warm Pool During the Lateglacial Driven by El Niño-like conditions"
- We also specified that we refer to the "East Asian" monsoon, where we previously used only "monsoon".
- We made the conclusions about the relevance of AMOC vs. orbital forcing clearer at the end of paragraph 4.4.
- We went through the manuscript once more and made a few improvements.
- The writing in the discussion section has been tightened and reorganized slightly to remove repetitions. This has not affected the overall content and information in the text.
- About the wording using "drier conditions coupled with…" on line 65: What we try to convey is that since speleothem and other proxies show drier conditions (unknown if seasonally or annual means), while other proxies and models show changes in ENSO/seasonality. We thus want to highlight that both are changing. We hope this is clearer by replacing the word "coupled" to "together".
- Updated the affiliation of one of the co-authors.